



# Sensitivity of Neoproterozoic Snowball-Earth inceptions to continental configuration, orbital geometry, and volcanism

Julius Eberhard[1,2], Oliver E. Bevan[3], Georg Feulner[1], Stefan Petri[1], Jeroen van Hunen[3], and James U.L. Baldini[3]

[1]Earth System Analysis, Potsdam Institute for Climate Impact Research, Member of the Leibniz Association, Potsdam, Germany
[2]Institute of Physics and Astronomy, University of Potsdam, Potsdam, Germany
[3]Department of Earth Sciences, Durham University, Durham, United Kingdom

**Correspondence:** Julius Eberhard (julius.eberhard@pik-potsdam.de) and Georg Feulner (feulner@pik-potsdam.de)

**Abstract.** The Cryogenian period (720–635 million years ago) in the Neoproterozoic era featured two phases of global or near-global ice cover, termed 'Snowball Earth'. Climate models of all kinds indicate that the inception of these phases must have occurred in the course of a self-amplifying ice–albedo feedback that forced the climate from a partially ice-covered to a Snowball state within a few years or decades. The maximum concentration of atmospheric carbon dioxide ($CO_2$) allowing such a

drastic shift depends on the choice of model, the boundary conditions prescribed in the model, and the amount of climatic variability. Many previous studies report values or ranges for this $CO_2$ threshold but typically test only very few different boundary conditions or exclude variability due to volcanism. Here we present a comprehensive sensitivity study determining the $CO_2$ threshold in different scenarios for the Cryogenian continental configuration, orbital geometry, and short-term volcanic cooling effects in a consistent model framework, using the climate model of intermediate complexity CLIMBER-$3\alpha$. The continental

configurations comprise two palaeogeographic reconstructions for each of both Snowball-Earth onsets, as well as two idealised configurations with either uniformly dispersed continents or a single polar supercontinent. Orbital geometries are sampled as multiple different combinations of the parameters obliquity, eccentricity, and argument of perihelion. For volcanic eruptions, we differentiate between single globally-homogeneous perturbations, single zonally-resolved perturbations, and random sequences of globally-homogeneous perturbations with realistic statistics. The $CO_2$ threshold lies between 10 and 250 ppm for

all simulations. While the idealised continental configurations span a difference of around 200 ppm for the threshold, the $CO_2$ thresholds for the continental reconstructions differ by only 20–40 ppm. Changes in orbital geometry account for variations in the $CO_2$ threshold by up to 32 ppm. The effects of volcanic perturbations largely depend on the orbital geometry and the corresponding structure of coexisting stable states. A very large peak reduction of net solar radiation by around 20 W m$^{-2}$ can shift the $CO_2$ threshold by the same order of magnitude as or less than the orbital geometry. Exceptionally large eruptions of

up to $-40$ W m$^{-2}$ shift the threshold by up to 50 ppm for one orbital configuration. Eruptions near the equator tend to, but do not always, cause larger shifts than eruptions at high latitudes. The effect of realistic eruption sequences is mostly determined by their largest events. In the presence of particularly intense small-magnitude volcanism, this effect can go beyond the ranges expected from single eruptions.



## 1 Introduction

Geologic evidence attests several episodes of global or near-global ice cover in Earth's past. The Neoproterozoic era, 1000–538.8 million years ago (Ma), features two such 'Snowball Earth' phases, commonly referred to as the Sturtian (c. 717–659 Ma) and the Marinoan glaciations (c. 650–635 Ma). Global occurrences of mostly synchronously deposited diamictites, dropstones, and tropical cap carbonates (e.g. Hoffman et al., 1998, 2017) render the transient but widespread existence of land ice virtually undisputed. Yet, a scientific debate revolves around the exact extent and time evolution of the ice cover. Regarding sea ice,

the suggestion of partially ice-covered tropical oceans (so-called Waterbelt, Slushball, Soft Snowball, or Jormungand states)—underpinned by qualitative or conceptual arguments (Kirschvink, 1992; Olcott et al., 2005; Moczydłowska, 2008; Allen and Etienne, 2008; Abbot et al., 2011)—opposes the 'Hard Snowball' hypothesis featuring a mainly closed cover of hundreds-metres-thick tropical sea ice (e.g. Hoffman et al., 2017). Views also differ on land-ice extent during Snowball glaciations, but strong geologic indications exist for cyclically shifting ice-sheet margins (Hoffman et al., 2017; Mitchell et al., 2021).

These findings have been interpreted with the help of climate models of differing complexity. Apart from providing climatic constraints on the geologic interpretations of Snowball conditions (reviewed in Hoffman et al., 2017), model solutions imply globally nearly synchronous onsets and globally synchronous ends of such phases. In particular, all climate models show Snowball inceptions and terminations as sudden and drastic shifts in phase space, caused by the positive ice–albedo feedback: given a small increase in the global albedo due to ice-area expansion, the absorbed solar radiation decreases and lowers the

surface temperature at and around the newly ice-covered latitudes. This cooling facilitates further ice expansion, again yielding a cooling effect. Above a certain ice extent, any ice expansion becomes self-amplifying and pushes the climate into a state of fully or mostly ice-covered oceans within a few years or decades. The inverse effect can explain the shift from a Snowball to an ice-free Earth.

Different frameworks exist within which to describe such shifts (e.g. Held and Suarez, 1974; Roe and Baker, 2010; Lucarini

et al., 2010; Lucarini and Bódai, 2017). Dynamical-systems theory, for example, provides the concepts of *trajectories*, along which a particular set of simulated variables moves with time, and *attractors*, phase-space regions within which the trajectories stay after some time with constant boundary conditions (e.g. Müller, 2022, their Chapter 4). Essentially, an attractor corresponds to the widely-used notion of an *equilibrium* or *steady* climate state. Upon approaching a Snowball Earth, a stable attractor of partially ice-covered states eventually becomes unstable and vanishes. This qualitative change in the model's phase

space, called bifurcation, leaves the system with a Snowball state as its nearest attractor. In fact, the Snowball and one or multiple other warmer attractors may coexist for a large range of radiative boundary conditions (exemplarily sketched in Figure 1), yielding a hysteresis upon changing these conditions.

Possible causes of Snowball inceptions and terminations can be investigated with climate models only to a certain degree. Regarding a Snowball termination, a gradual buildup of the atmospheric carbon-dioxide ($CO_2$) concentration through volcanism,

probably facilitated by reduced weathering in the presence of a global ice cover (Walker et al., 1981) and maybe complemented by volcanic dust deposition on the ice (e.g. Abbot and Pierrehumbert, 2010; Abbot and Halevy, 2010; Le Hir et al., 2010), is usually considered as a likely cause (e.g. Hoffman et al., 2017). Going from a partially ice-covered to a Snowball Earth, a



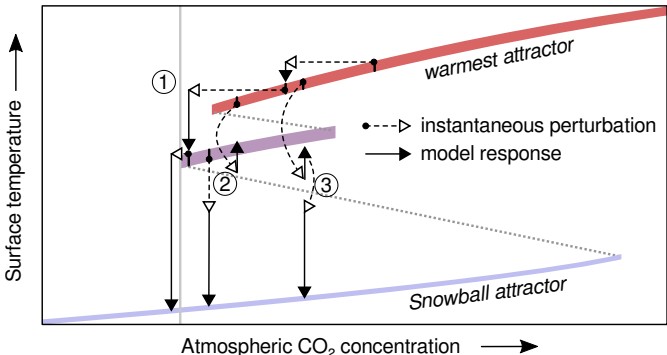

**Figure 1.** Schematic sketch of two non- or partially ice-covered attractors and a Snowball attractor (coloured stripes), projected onto the surface-temperature axis, for a range of atmospheric $CO_2$ concentrations. Dotted grey stripes represent intermediate and mostly unstable attractors separating the basins of attraction. In our study, Snowball inceptions follow from either ① a stepwise reduction of $CO_2$ ('B-tipping', Ashwin et al., 2012), ② a single transient perturbation of surface temperature, or ③ a fast sequence of transient perturbations (in both cases 'S-tipping', Halekotte and Feudel, 2020). For ①, the inception occurs below the equilibrium threshold (vertical grey line). ② and ③ yield Snowballs below perturbed thresholds.

runaway ice–albedo feedback can be triggered either by a significant increase in land-ice extent or by sustained local or global cooling. The long-term effect of dynamic ice sheets on a Snowball inception has been tested only sparsely (e.g. Hyde et al.,
2000). In contrast, a large number of studies discuss potential causes for sustained cooling or a high Proterozoic susceptibility to glaciation, including the lower solar luminosity (Feulner, 2012; Feulner et al., 2023), a modification of atmospheric-$CO_2$ drawdown due to emerging land plants around 470 Ma (Brady and Carroll, 1994; Pierrehumbert, 2010, their Chapter 8) or carbonate-precipitating plankton (Ridgwell et al., 2003), variations in atmospheric methane (Pierrehumbert et al., 2011), increased reflectivity of clouds (Feulner et al., 2015), the $CO_2$ sequestration resulting from large tropical continents (Hoffman
and Schrag, 2002), or the breakup of the supercontinent Rodinia (Goddéris et al., 2003).

Most modelling studies scan for critical radiative conditions, here called equilibrium thresholds, at which the coldest non-Snowball attractor vanishes (scenario 1 in Figure 1; a type of 'bifurcation-induced [B-]tipping' as classified by Ashwin et al., 2012). These thresholds are mostly specified as critical values of the solar constant $S_0$ or the atmospheric $CO_2$ concentration, which can vary considerably between different models (Yang et al., 2012a, b, c; Feulner et al., 2023) and depend on different
boundary conditions, e.g. ozone concentration (Jenkins, 1999), the degree of desertification (Hoffman et al., 2017), and ocean salinity (Olson et al., 2022).

Another boundary condition on which the bifurcation depends is the distribution of continents on Earth's surface. The continental configuration primarily affects the planetary energy balance via the surface albedo, its influence on the presence of continental ice and snow, and changes in ocean-circulation patterns. In the context of the Snowball inceptions during the Neo-
proterozoic, this has been investigated in a number of studies (e.g. Lewis et al., 2003; Voigt et al., 2011; Liu et al., 2013), using either idealised continental configurations, reconstructions, or the difference between Neoproterozoic and present-day land–





ocean distributions. Recently, Merdith et al. (2021) published a new reconstruction of Neoproterozoic continental distributions which has not been used in quantifications of Snowball inceptions yet.

One further point we would like to emphasise is that the search for equilibrium thresholds neglects the fact that Earth's climate is subject to additional forcings causing natural climate variability on a wide range of timescales. For example, periodic changes in Earth's orbital parameters can generate climate variations on timescales of ~$10^4$–$10^5$ yr, as fluctuations in eccentricity, obliquity, and the timing of the perihelion vary the geographic and seasonal distribution of the incoming solar radiation (Milanković, 1941; Donnadieu et al., 2002). Lewis et al. (2003) found marked differences in climate for two selected orbital configurations under Neoproterozoic boundary conditions, but a more complete range of Earth's orbital parameters still has to be investigated.

On even shorter timescales of ~1–$10^2$ years, single or sequential large volcanic eruptions can cool Earth's climate due to the formation of stratospheric sulfate aerosols (Robock, 2000). In principle, such perturbations are able to trigger a Snowball inception below some 'perturbed threshold', a $CO_2$ concentration above the equilibrium threshold (scenarios 2 and 3 in Figure 1; types of 'shock-induced [S-]tipping' from single or sequential large eruptions, Halekotte and Feudel, 2020). The potential role of volcanic eruptions associated with the Franklin Large Igneous Province in triggering the Sturtian glaciation has been investigated by Macdonald and Wordsworth (2017). They argue that the cool climate states of the Neoproterozoic are particularly sensitive to the effects of volcanic eruptions due to the lower tropopause height, resulting in a higher sulfate flux to the stratosphere. In a series of simulations, Gupta et al. (2019) explored the ability of single and sequential eruptions triggering transitions between attractors. A more systematic study of the effect of volcanic eruptions of different magnitudes on the Sturtian and Marinoan Snowball-inception thresholds in a coupled climate model is still lacking, however.

Here, we study both the sensitivity of the equilibrium threshold to continental configuration and orbital geometry, i.e. the combination of orbital parameters, and that of the perturbed threshold to volcanism for both Neoproterozoic Snowball inceptions in a coupled climate model of intermediate complexity (Montoya et al., 2005). In terms of continental configuration, we test the recent Merdith et al. (2021) reconstructions against earlier ones by Li et al. (2013) as well as more idealised end-member configurations of widely dispersed landmasses and a supercontinent centred at the South Pole, respectively. The sensitivity of the Snowball inception with respect to orbital parameters is investigated by sampling the parameter space of Earth-like orbital configurations. Due to the long timescale of Milanković-forcing variations (Milanković, 1941) and the prohibitive computational demands for transient simulations, we do this in terms of equilibrium simulations. Finally, we quantify the perturbed thresholds due to single volcanic eruptions of different magnitude and with globally-homogeneous or zonally-resolved aerosol loads as well as the effect of sequential eruptions with realistic statistics.

This paper is organised as follows: in Section 2, we describe the model used for our investigation as well as the boundary conditions and sensitivity experiments. The following sections focus on the sensitivity of the Neoproterozic climate system and the Snowball inception to continental configuration (Section 3), orbital parameters (Section 4), and volcanic eruptions (Section 5) and discuss the results in the context of related work. Finally, Section 6 summarises our findings.



## 2 Model, simulation setup, analytical methods

### 2.1 Model

As in our earlier investigations of Neoproterozoic glaciations (Feulner and Kienert, 2014; Feulner et al., 2015), we use the relatively fast intermediate-complexity model CLIMBER-3$\alpha$ (Montoya et al., 2005) to be able to run a large number of simulations. CLIMBER-3$\alpha$ consists of (1) an improved version of the ocean general circulation model MOM3 (Pacanowski and Griffies, 2000; Montoya et al., 2005; Hofmann and Morales Maqueda, 2006) run at a coarse horizontal resolution of $3.75° \times 3.75°$ with 24 vertical layers, (2) the sea-ice model ISIS (Fichefet and Morales Maqueda, 1997; Hunke and Dukowicz, 1997) operated at the same horizontal resolution and capturing both the thermodynamics and dynamics of sea ice, and (3) the fast statistical–dynamical atmosphere model POTSDAM-2 (Petoukhov et al., 2000) with grid cells measuring $22.5°$ in longitude and $7.5°$ in latitude. CLIMBER-3$\alpha$ does not support Waterbelt states with narrow regions of open ocean around the equator. However, it has been shown to support multiple stable cold climate states with extensive sea-ice cover in an aquaplanet setup (Feulner et al., 2023).

The albedo parametrisations of ice- and snow-covered surfaces can differ significantly between climate models and are known to have crucial effects on Snowball inceptions (e.g. Pierrehumbert et al., 2011; Yang et al., 2012c). In our model setup (based on Feulner and Kienert, 2014), bare sea ice has a clear-sky albedo of up to 0.32 for infrared and up to 0.62 for visible and ultraviolet radiation. Snow-covered sea ice has maximum clear-sky albedos of 0.57 and 0.87. For overcast conditions, sea-ice albedos are raised by 0.06. Snow on land has maximum albedos of 0.79 (infrared) and 0.97 (visible/ultraviolet) for clear sky, which are lowered to 0.65 and 0.95, respectively, for overcast sky. Snow albedos decrease depending on snow age, snow temperature, dust cover, and the solar zenith angle.

In addition to the coarse spatial resolution, potential limitations of the model mostly result from the simplified atmosphere component, which statistically captures the large-scale atmospheric circulation patterns rather than simulating weather, and the absence of ice sheets. Although the large-scale atmospheric circulation dynamically responds to changes in the boundary conditions, some aspects are fixed, in particular the annual-mean width of the Hadley circulation cells (Petoukhov et al., 2000). Since CLIMBER-3$\alpha$ does not include interactive ice sheets, we run the model without any land ice. Despite these limitations, results for the Snowball-inception threshold in the Neoproterozoic fall well within the range derived from more complex models (Feulner et al., 2023). We will discuss the potential impact of model limitations on our results in Section 6.

### 2.2 Simulation setup

According to our research question, we proceed along three different tracks, involving simulations with either (1) different continental configurations (Section 2.2.1), (2) different orbital geometries (Section 2.2.2), or (3) different scenarios of volcanic activity (Sections 2.2.3 to 2.2.5).

Simulations in (1) and (2) are designed as sets of unperturbed simulations. In the context of this study, 'unperturbed' means that we leave the solar constant, the continental configuration, and the orbital geometry unchanged throughout each run and let the model approach an attractor.





Volcanic eruptions are emulated as prescribed transient perturbations of the incoming solar radiation, either globally homogeneous or zonally resolved. Starting from single global perturbations of different magnitudes at three different orbital
geometries (Section 2.2.3), we proceed investigating the effects of different latitudes of eruption (Section 2.2.4) and sequences of eruptions (Section 2.2.5) with reference to the single global eruptions at one particular orbital geometry.

### 2.2.1   Sampling continental configurations

For each of the two Neoproterozoic Snowball-Earth onsets—the Sturtian onset (taken here as 720 Ma) and the Marinoan (here 650 Ma)—four different continental configurations are applied as boundary conditions: (1) two reconstructions and (2)
two idealised configurations. Figure 2 shows the distribution of continents and the bathymetry in the different configurations along with the zonally-averaged land fraction. The idea behind this sampling is (1) to obtain an estimate of the uncertainty induced by using different realistic palaeogeographies and (2) to roughly cover the largest possible uncertainty of global-mean temperatures caused by the distribution of continents.

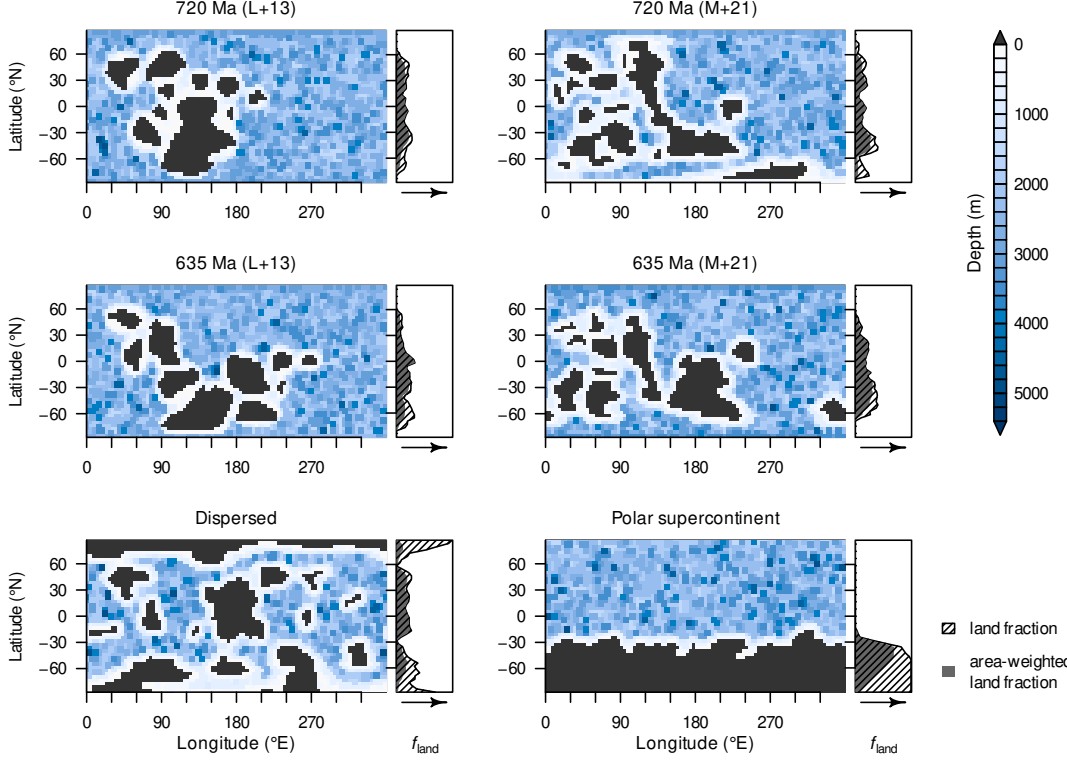

**Figure 2.** Distribution of exposed continental surface, ocean bathymetry, and zonally averaged land fraction for the different continental configurations. Age specifications (720 and 635 Ma) refer to the time of the reconstructions, L+13 denotes reconstructions based on Li et al. (2013), M+21 those based on Merdith et al. (2021). Land fractions are shown as simple zonal averages (hatched) and zonal averages weighted by the actual area of the latitudinal bands (grey).





The reconstructions are based on published datasets from Li et al. (2013) and Merdith et al. (2021), dated to 720 Ma for

the Sturtian and 635 Ma for the Marinoan Snowball onset. (Note, however, that we perform Marinoan simulations with a solar constant at 650 Ma.) These reconstructions rely upon narrowly-constrained palaeomagnetic datasets derived from cratonic lithologies, as well as plate kinematics and tectonostratigraphic correlations. The more refined methods applied in the Merdith et al. (2021) palaeogeography allow for the trajectories of large terranes to also be projected through geologic history, producing a more robust model foundation during a time of hypothesised metamorphism following the breakup of the supercontinent

Rodinia (Merdith et al., 2017). For the remainder of the paper, the reconstructions will be referred to as L+13 for the ones based on Li et al. (2013) and M+21 for those based on Merdith et al. (2021).

The land–ocean masks based on the reconstructions are re-gridded to the ocean-model grid with a horizontal resolution of $3.75° \times 3.75°$. Some grid cells have to be manually adjusted, for example to avoid isolated ocean cells within continents or narrow ocean channels. For the L+13 reconstruction, continental shelves and slopes with a typical width of one ocean grid cell

each are assigned by hand around all land areas (Feulner and Kienert, 2014). For the M+21 reconstruction and the idealised configurations, we use an automatic procedure with shelves and slopes with a width of 500 km each. Ocean depth levels are randomly assigned and fall within the range 100–265 m for continental shelves, 349–993 m for continental slopes, and 1243–4987 m for the abyss. Land cells are set to an elevation of either constantly 200 m or randomly distributed between 200 and 400 m. Land-surface types are assumed to be bare soil, unless in case of snow cover.

The idealised continental configurations are constructed by shifting continental plates of the 635-Ma reconstruction by Merdith et al. (2021), using the 'GPlates' program (Müller et al., 2018) and subsequent manual adjustments as described above. We construct a 'dispersed' configuration of small continents distributed evenly over the surface and a 'polar supercontinent' configuration, where the single continent is the only landmass and is centred at the South Pole of the model grid. Depth levels of continental shelves, slopes, and deep-ocean cells are assigned in the same manner as for the M+21 reconstruction described

above.

The use of different sources and the manual adjustments come with variations in the exposed continental-surface area. It is lower for the reconstructions, with the 720-Ma land fraction (L+13: 16.8 %, M+21: 17.1 %) being smaller than at 635 Ma (L+13: 17.4 %, M+21: 20.0 %). The idealised dispersed configuration takes a land fraction of 20.8 %, the supercontinent 24.1 %.

Table 1 provides an overview of the simulations with differing continental configurations. We use the solar constant $S_0 = 1283\,\mathrm{W\,m^{-2}}$ for the Sturtian and $1290\,\mathrm{W\,m^{-2}}$ for the Marinoan onset. Both values are derived from a standard solar model (Bahcall et al., 2001), using a present-day value of $1361\,\mathrm{W\,m^{-2}}$. For the orbital geometry we select an obliquity of $23.5°$ and zero eccentricity. All unperturbed runs are initialised in a warm, ice- and snow-free state with a global-mean surface-air temperature of around $12\,°\mathrm{C}$.

In the simulations aiming at a comparison of the energy-balance effects from differing continental distributions, we run the model with an atmospheric $CO_2$ concentration of 250 ppm for the Sturtian and 200 ppm for the Marinoan conditions. Non-$CO_2$ greenhouse gases are not considered in our simulations. Scanning for the $CO_2$ threshold is done by letting the model approach equilibrium for various $CO_2$ concentrations in steps of 10 ppm.



**Table 1.** Overview of the simulations regarding the sensitivity to continental distribution, differentiating between those aimed at the temperature sensitivity at fixed $CO_2$ concentration and those aimed at the $CO_2$ threshold for Snowball inception. All simulations are initialised in a warmer climate than the shown equilibrium states and run with an orbital geometry of 23.5° obliquity and zero eccentricity. $S_0$ is the solar constant. L+13 denotes reconstructions based on Li et al. (2013), M+21 those based on Merdith et al. (2021).

| Sensitivity of ... | $S_0$ (W m$^{-2}$) | Continents | $CO_2$ concentration (ppm) |
|---|---|---|---|
| | 1283 (720 Ma) | 720 Ma (L+13) | 250 |
| | | 720 Ma (M+21) | 250 |
| | | Dispersed | 250 |
| | | Polar supercontinent | 250 |
| ...equilibrium surface-air temperature | 1290 (650 Ma) | 635 Ma (L+13) | 200 |
| | | 635 Ma (M+21) | 200 |
| | | Dispersed | 200 |
| | | Polar supercontinent | 200 |
| | 1283 (720 Ma) | 720 Ma (L+13) | 137, 138 |
| | | 720 Ma (M+21) | 90, 100 |
| | | Dispersed | 240, 250 |
| | | Polar supercontinent | 30, 40 |
| ...equilibrium Snowball threshold in $CO_2$ | 1290 (650 Ma) | 635 Ma (L+13) | 170, 180 |
| | | 635 Ma (M+21) | 160, 170 |
| | | Dispersed | 190, 200 |
| | | Polar supercontinent | 10, 20 |

### 2.2.2 Sampling orbital geometries

In CLIMBER-3$\alpha$, the incoming top-of-atmosphere solar radiation is determined by three orbital parameters: obliquity $\varepsilon$, eccentricity $e$, and argument of perihelion $\omega$. $\omega = 0°$ corresponds to perihelion at the September (southward) equinox and $\omega = 90°$ to that at the December (southern) solstice. We sample the $\varepsilon$–$e$–$\omega$ parameter space by selecting the coordinates $\varepsilon = (22°, 23.5°, 24.5°)$, $e = (0, 0.03, 0.069)$, $\omega = (0°, 45°, 90°, 135°, 180°, 225°, 270°, 315°)$ and running the model for selected, physically-distinguishable combinations thereof. (Note that for $e = 0$, the different values of $\omega$ are physically indistinguishable.) The

ranges of $\varepsilon$ and $e$ are confined by computations of Earth's orbital variations during the past 250 million years (Laskar et al., 2004). The climate of these simulations is initialised in the same way as for the set of simulations described above.

An overview of the simulations with varying orbital geometries is given in Table 2. For all 51 distinguishable configurations, we run the model at a $CO_2$ concentration of 150 ppm in order to compare the effects of orbital geometry on the surface-air temperature. Since these effects are significantly smaller than those of the continental configurations, we perform a finer





scanning for the $CO_2$ thresholds at nine different orbital geometries: After letting the model approach a steady state, which usually takes around 1000–3000 yr, we reduce $CO_2$ by 1 ppm—or, well above the threshold, more—, and repeat this procedure until we have a Snowball inception in each simulation. The continents and solar constant always correspond to the Sturtian onset (720 Ma), using the respective L+13 continental reconstruction.

**Table 2.** Overview of the simulations regarding the sensitivity to orbital geometry, differentiating between those aimed at the temperature sensitivity at fixed $CO_2$ concentration and those aimed at the $CO_2$ threshold for Snowball inception. All simulations are initialised in a warmer climate than the shown equilibrium states, run at a solar constant of $1283\,\mathrm{W\,m}^{-2}$, and use the Sturtian L+13 palaeogeography. $\varepsilon$ is obliquity, $e$ eccentricity, $\omega$ argument of perihelion.

| Sensitivity of . . . | $\varepsilon$ (°) | $e$ | $\omega$ (°) | $CO_2$ concentration (ppm) |
|---|---|---|---|---|
| | | 0.000 | 0 | 150 |
| | 22.0 | 0.030 | 0, 45, 90, . . . , 315 | 150 |
| | | 0.069 | 0, 45, 90, . . . , 315 | 150 |
| | | 0.000 | 0 | 150 |
| . . . equilibrium surface-air temperature | 23.5 | 0.030 | 0, 45, 90, . . . , 315 | 150 |
| | | 0.069 | 0, 45, 90, . . . , 315 | 150 |
| | | 0.000 | 0 | 150 |
| | 24.5 | 0.030 | 0, 45, 90, . . . , 315 | 150 |
| | | 0.069 | 0, 45, 90, . . . , 315 | 150 |
| | | 0.000 | 0 | 150, 130, 124, 123 |
| | 22.0 | 0.069 | 90 | 150, 120, 113, 112, 111, 110 |
| | | 0.069 | 270 | 150, 115, 110, 109, 108 |
| | | 0.000 | 0 | 150, 140, 138, 137 |
| . . . equilibrium Snowball threshold in $CO_2$ | 23.5 | 0.069 | 90 | 150, 115, 114, 113 |
| | | 0.069 | 270 | 150, 120, 113, 112 |
| | | 0.000 | 0 | 150, 141, 140 |
| | 24.5 | 0.069 | 90 | 150, 125, 119, 118 |
| | | 0.069 | 270 | 150, 120, 116, 115 |

### 2.2.3  Sampling single global volcanic perturbations

We implement globally-averaged volcanic perturbations as annually-resolved transient variations in the solar constant. The radiative perturbation resulting from a single explosive eruption is idealised as follows: in the calendar year *after* the eruption, the perturbation reaches a peak forcing, followed by an exponential decrease $\sim \mathrm{e}^{-\lambda t_i}$ with the successive calendar years $t_i$ and the decay rate $\lambda$, having a value of $1\,\mathrm{yr}^{-1}$. In the calendar year *of* the eruption the forcing is half the peak forcing, representing





a linear buildup due to sulfate aerosols being injected into the stratosphere. This form of the perturbation, including the value
of $\lambda$, is commonly used in the literature (e.g. Ammann et al., 2003; Gao et al., 2008; Metzner et al., 2014).

With the aim of sampling the peak forcing realistically, we construct our idealised scenarios following ice-core-based recon-
structions of the net-solar-radiative forcing $\Delta S_{\mathrm{net}}(t)$ due to volcanism during the years 850–2000 Common Era (CE; Schmidt
et al., 2012, based on Gao et al., 2008, and Crowley et al., 2008) as well as model-based forcings of the ~74 000-a Toba (Timm-
reck et al., 2010) and 2.1-Ma Yellowstone eruptions (Segschneider et al., 2013). The annual- and global-mean peak forcings
from these reconstructions cover a range of up to $-20\,\mathrm{W\,m^{-2}}$. We define our single-eruption scenarios using a nominal reduc-
tion of the global-mean net solar radiation, here called 'nominal volcanic forcing' (NVF), with peaks of $-2$, $-5$, $-10$, $-20$,
$-30$, $-40\,\mathrm{W\,m^{-2}}$, thus also allowing for what we consider extreme and very rare perturbations beyond the reconstructions.
Note that some previously-applied (Jones et al., 2005; Robock et al., 2009; Gupta et al., 2019) estimates of the ~74 000-a
Toba eruption exceed a peak forcing of $-100\,\mathrm{W\,m^{-2}}$ but have been criticised on physical grounds as being much too strong
(Timmreck et al., 2010, 2012).

The translation of a given NVF to a change in the solar constant results from a basic consideration of the top-of-atmosphere
shortwave-radiation balance: the daily- and zonal-mean net solar radiation is $S_{\mathrm{net}}(\phi) = A(\phi)g(\phi,\phi_{\mathrm{z}})S_0$, where $\phi$ is the latitude,
$A$ the planetary coalbedo ($1-$albedo), $g$ a factor comprising geometric effects of Earth and its orbit, and $\phi_{\mathrm{z}}$ the latitude at which
the Sun reaches the zenith at noon (e.g. Peixoto and Oort, 1992, their Section 6.3.2). For the annual and global mean, denoted
by an overline, we approximate $\overline{A(\phi)g(\phi,\phi_{\mathrm{z}})}$ by $A_{\mathrm{eff}}\overline{g}$, where $A_{\mathrm{eff}}$ is a geometry- and radiation-weighted effective global
coalbedo and $\overline{g} = (4\sqrt{1-e^2})^{-1}$ (Berger and Loutre, 1994). We use the present-day global coalbedo of 0.7 (Goode et al.,
2001) as an estimate for $A_{\mathrm{eff}}$. Thus we translate the NVF into the solar-constant forcing $\Delta S_0$ as

$$\Delta S_0(t_i) = \frac{4\sqrt{1-e^2}}{0.7}\mathrm{NVF}(t_i)$$

with the successive calendar years $t_i$.

Three comments on the construction of the volcanic forcing are appropriate at this point. First, the volcanic forcing is called
'nominal' here because by this construction each eruption corresponds to a reduction of the solar constant of around 5.7 NVF
for the used eccentricity range, independent of the actual climate state. This means that, while the NVF represents the volcanic
effects in a Quaternary or modern climate, the actual reduction in the net solar flux is somewhat smaller in a colder climate
with a lower coalbedo. For the sake of simplicity and comparability with other studies, however, we continue to specify the
eruption scenarios using the NVF.

Second, the use of annual reductions in the solar constant implies volcanic eruptions always starting at the beginning of a
calendar year, i.e. in northern-hemispheric winter.

Third, Macdonald and Wordsworth (2017) argued on basis of a vertical atmospheric and volcanic-plume model that volcanic
eruptions can have a particularly strong impact on Snowball inceptions because the tropopause, the lower boundary of the
stratosphere, is generally lower in cooler climates. We do not explicitly include such an effect here.





All perturbed runs, including those described in the sections below, have Sturtian boundary conditions, making use of the respective L+13 palaeogeography. Only the single global perturbations are tested for different orbital geometries, which are $\{\varepsilon = 22°, e = 0\}$, $\{\varepsilon = 23.5°, e = 0\}$, and $\{\varepsilon = 24.5°, e = 0.069, \omega = 270°\}$. See Table 3 for an overview of these simulations.

**Table 3.** Overview of the simulations regarding the sensitivity of the $CO_2$ threshold for Snowball inception to single global volcanic perturbations. All simulations are run at a solar constant of $1283\,\mathrm{W\,m^{-2}}$ and use the L+13 palaeogeography for 720 Ma. $\varepsilon$ is obliquity, $e$ eccentricity, $\omega$ argument of perihelion.

| $\varepsilon$ (°) | $e$ | $\omega$ (°) | Initial partially ice-covered attractor | Nominal volcanic peak forcing ($\mathrm{W\,m^{-2}}$) | $CO_2$ concentration (ppm) |
|---|---|---|---|---|---|
| | | | Warmest | −2 | 124 |
| | | | | −5 | 126, 125 |
| | | | | −10, −20, −30 | 133, 132 |
| | | | | −40 | 138, 137 |
| | | | Coldest | −2 | 125 |
| 22.0 | 0.000 | 0 | | −5 | 127, 126 |
| | | | | −10 | 133, 132 |
| | | | | −20 | 151, 150 |
| | | | | −30 | 160, 159 |
| | | | | −40 | 173, 172 |
| 23.5 | 0.000 | 0 | Warmest | −2, −5, −10 | 139, 138 |
| | | | | −20, −30, −40 | 140, 139 |
| | | | Coldest | −2, −5 | 139 |
| | | | | −10 | 141, 140 |
| | | | | −20, −30, −40 | 140, 139 |
| 24.5 | 0.069 | 270 | Warmest | −2 | 116 |
| | | | | −5 | 117, 116 |
| | | | | −10 | 118, 117 |
| | | | | −20 | 129, 128 |
| | | | | −30 | 136, 135 |
| | | | | −40 | 143, 142 |
| | | | Coldest | −2, −5 | 117 |
| | | | | −10 | 120, 119 |
| | | | | −20 | 130, 129 |
| | | | | −30 | 135, 134 |
| | | | | −40 | 147, 146 |



### 2.2.4 Sampling single zonal volcanic perturbations

To emulate the latitudinally-resolved effects of large volcanic eruptions, we prescribe a transient zonal distribution of aerosol optical depth (AOD) yielding a proportional reduction of the solar constant at each latitude band. The aerosol time series are taken from Ammann et al. (2003), who estimated the monthly- and zonally-resolved AOD for historical eruptions between 1890 and 1999 CE based on a simple model of sulfate-aerosol transport in the stratosphere. We exemplarily select the periods of 6–9 yr following three eruptions at different latitudes: 1902 Gagxanul / Santa María (at $15°$ N), 1932 Quizapú / Cerro Azul

($36°$ S), and 1912 Novarupta ($58°$ N). We interpolate these data to our atmospheric grid and prolong them to cover 50 yr, assuming an exponential decay with a rate of $1\,\mathrm{yr}^{-1}$ after the last available values (Figure A1). Additionally, we flip the data at the equator to eventually have AOD time series for eruptions at $15°$, $36°$, and $58°$ in both hemispheres.

In order to produce comparable results, we scale the AOD data for each peak forcing in such a way that the 50-yr-integrated, global-mean reduction of the solar constant is equal to the 50-yr integral of $\Delta S_0(t)$ in the respective global volcanic perturba-

tion as described above. Regarding their effects on the model response, the only differences to the global perturbations are the finer spatial (latitudinal) and temporal (monthly) resolution. Note in particular that, as in the global case, eruptions always start in January.

All zonal perturbations are initialised on both the warmest and the coldest ice-free attractors for the orbital geometry $\{\varepsilon = 22°,\ e = 0\}$. See Table 4 for an overview.

### 2.2.5 Sampling sequences of global volcanic perturbations


We generate sequences of global volcanic perturbations along the lines of Ammann and Naveau (2010), who construct such annual-mean time series based on two processes, one purely stochastic and one purely deterministic:

First, the occurrence of explosive eruptions ejecting sulfate aerosols into the stratosphere is modelled as a stochastic annual time series $b_p(t_i)$ consisting of binary values ('no eruption' or 'eruption', i.e. a Bernoulli process), where 'eruption' has the

constant probability $p$ and 'no eruption' the probability $1 - p$. Each such eruption is then assigned a nominal volcanic peak forcing $\Pi(t_i)$ randomly drawn from a generalised Pareto distribution with probability density

$$g_{\beta,\xi,u}(x) = \frac{1}{\beta}\left(1 + \frac{\xi}{\beta}\frac{u-x}{1\,\mathrm{W\,m}^{-2}}\right)^{-1/\xi - 1}$$

where $\beta > 0$, $\xi$ are real numbers; $u$, $x$ are given in $\mathrm{W\,m}^{-2}$ and ensure that $u > x$ and the term in parentheses is positive. This function asymptotically describes excesses over large thresholds of many probability distributions (Embrechts et al.,

1997), including the peak radiative forcing of large volcanic eruptions (Naveau and Ammann, 2005). Note that in our case, the distribution describes negative events $x$ beyond a negative threshold $u$. The nominal volcanic forcing, i.e. reduction of the net solar radiation, is translated into a reduction of the solar constant in the same way as described in Section 2.2.3. The complete




first process, describing subsequent peaks $\Sigma$ in the perturbation of the solar constant, therefore reads

$$\Sigma(t_i) \stackrel{\mathrm{d}}{=} \begin{cases} \frac{4\sqrt{1-e^2}}{0.7}\Pi(t_i) \text{ with } \mathrm{Prob}(\Pi(t_i) \in [x+\mathrm{d}x]) = \mathrm{d}x\, g_{\beta,\xi,u}(x) & \text{if } b_p(t_i) = \text{eruption}, \\ 0 & \text{if } b_p(t_i) = \text{no eruption}, \end{cases}$$

where 'd' over the equality sign implies equality in distribution.

Second, each peak is followed by an exponential decay as in our idealised single perturbations, described by the deterministic process $\Delta S_0(t_i) = \Delta S_0(t_{i-1})/\mathrm{e}$.

Both processes are combined, yielding the max-autoregressive model

$$\Delta S_0(t_i) = -\max\{-\Delta S_0(t_{i-1})/\mathrm{e}, -\Sigma(t_i)\} \quad \text{with} \quad \Delta S_0(t_0) = 0.$$

Taking in each year the maximum instead of the sum ensures that the effective distribution of peak forcing follows the distribution $g_{\beta,\xi,u}$ even for overlapping events.

The values of the parameters $p$, $\beta$, $\xi$, and $u$—characterising the frequency of eruptions and distribution of peak forcings—define our scenarios. Roughly oriented towards the assumptions and parameter-fitting results by Ammann and Naveau (2010), we define the reference scenario as $\{p = 0.1,\ \beta = 1,\ \xi = 0.3,\ u = 1.5\,\mathrm{W\,m^{-2}}\}$, reproducing volcanic-forcing statistics of the

past ~1000 yr by and large. Based on this scenario, we subsequently change one parameter at a time in order to define five more scenarios. They comprise raising the eruption probability $p$ to 0.3, varying the shape parameter $\beta$ between 0.5 and 1.5 (a slightly wider range than in Ammann and Naveau, 2010), and shifting the threshold $u$ to 0 and $-3\,\mathrm{W\,m^{-2}}$ (as in Ammann and Naveau, 2010). Tripling the probability of eruptions with a relevant radiative effect is an attempt to represent periods of more frequent volcanic activity, e.g. during the formation of large igneous provinces (LIPs) as the Franklin LIP related to

the Sturtian Snowball inception by Macdonald and Wordsworth (2017). While $u = 0\,\mathrm{W\,m^{-2}}$ should be seen as a theoretical case only, because the generalised Pareto distribution is usually applied for nonzero thresholds (Naveau and Ammann, 2005), $u = -3\,\mathrm{W\,m^{-2}}$ is chosen as the other limiting case since already some effects can be seen from peak forcings of $-2\,\mathrm{W\,m^{-2}}$, as reported in Section 5.1. Changing $\xi$ within or slightly beyond the range reported in Ammann and Naveau (2010) results in distributions very similar to some scenarios obtained from varying $\beta$ and $u$. We therefore chose to leave it fixed at $\xi = 0.3$ for

all scenarios, yielding a Fréchet-type distribution with heavy tails (Embrechts et al., 1997).

Figure 3A shows the five different parametrisations of $g_{\beta,\xi,u}$. In order to make the interpretation of results easier, we compute the pseudorandom sequences with a length of 5000 yr for all different scenarios based on the same seed, i.e. initial algorithm state. By plotting the peak forcings of the reference scenario together with the peak forcings from other scenarios for each time step (Figure 3B), we see this relationship as perfect correlations between the time series. Moreover, we can clearly characterise

the effects of changing $\beta$ and $u$: through the scaling effect of $\beta$, the scenarios yield dramatically different forcings for the rarest events, while $u$ shifts the forcings homogeneously and thereby has the strongest effects on the most frequent eruptions, i.e. the 'background noise'. Due to the construction, the scenario with increased eruption probability $p$ contains almost all individual events, including their magnitude, from the reference case and adds many additional events (Figures 3C, D). The background noise is strongly intensified, leading to a nearly permanent effect of volcanoes on the radiative balance. Additionally, this

scenario contains a peak forcing (at around year 2000 in Figure 3D) going beyond the largest forcing in the reference scenario.



All simulations are initialised on the warmest partially ice-covered attractors with orbital geometry $\{\varepsilon = 22°, e = 0\}$. The reason for choosing only the warmest attractors is that in many simulations the sequential perturbations drive the model from the warmest to the coldest attractor as an intermediate state, in which cases the thresholds are independent from the initial state. In other words, the volcanic 'noise' seems to connect the multiple stable states and push the model towards the coldest one. After applying the volcanic forcing for 5000 yr, we let the model run without perturbations for 1000 yr. Table 5 summarises the simulations with sequential volcanic perturbations.

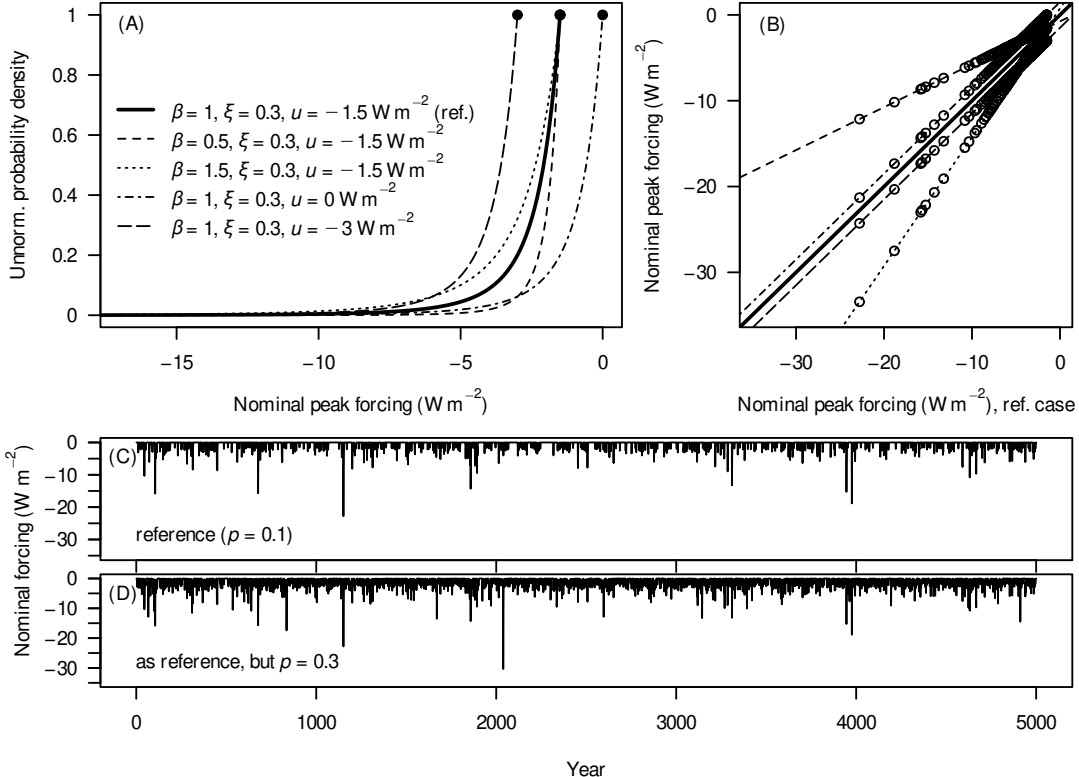

**Figure 3.** A: different forms of the generalised Pareto distribution determining the nominal peak forcing in the max-autoregressive model, displayed as unnormalised probability-density functions, i.e. having a maximum value of 1. Dots denote the upper limits of each function's domain and range. E.g. the function with $u = -3\,\mathrm{W\,m}^{-2}$ allows for peak forcings at or below $-3\,\mathrm{W\,m}^{-2}$ only, while all functions are unbounded from below. B: nominal peak volcanic forcing in the simulations with each of the distributions from panel A, compared to the reference scenario (diagonal line) at each time step of the generated volcanic time series. C: nominal volcanic forcing in the reference scenario. D: same as panel C but for the scenario with increased eruption probability.

## 2.3 Estimating energy-balance contributions to surface-temperature differences

Following an idea first deployed by Heinemann et al. (2009) and later used by Voigt (2010), Voigt et al. (2011), and Liu et al. (2013), we can divide the zonal-mean surface-temperature differences between two equilibrium states into contributions





associated with terms of the zonal energy balance. Similarly to the classic energy-balance models, the instantaneous zonal-mean energy balance of an attractor state reads

$$S_{\text{net}}(\phi,t) - L(\phi,t) - \nabla_\phi F(\phi,t) = \delta(\phi,t), \tag{1}$$

where $S_{\text{net}}$ is the net solar (shortwave) flux, $L$ the net outgoing terrestrial (longwave) flux, $\nabla_\phi F$ the divergence of meridional sensible and latent heat fluxes, and $\delta$ represents the fluctuating energy imbalance. All variables represent zonal averages. Parametrising the outgoing thermal radiation as

$$L(\phi,t) = \sigma \epsilon T^4(\phi,t),$$

where $\sigma \approx 5.67 \cdot 10^{-8}\,\mathrm{W\,m^{-2}\,K^{-4}}$ is the Stefan–Boltzmann constant, $\epsilon$ the effective emissivity, and $T$ the surface temperature, and taking the long-term mean ($\langle\cdot\rangle$) yields

$$\langle S_{\text{net}}\rangle(\phi) - \sigma\langle\epsilon T^4\rangle(\phi) - \langle\nabla_\phi F\rangle(\phi) = 0,$$

where, by definition of the attractor and assuming ergodicity, $\langle\delta\rangle(\phi) = 0$ for all $\phi$. If we approximate $\langle\epsilon T^4\rangle \approx \langle\epsilon\rangle\langle T\rangle^4$, we obtain for the long-term- and zonal-mean surface temperature

$$\langle T\rangle \approx \left[\frac{1}{\sigma\langle\epsilon\rangle}\left(\langle S_{\text{net}}\rangle - \langle\nabla_\phi F\rangle\right)\right]^{1/4} \tag{2}$$

In the model experiments, the single terms are diagnosed as

$$S_{\text{net}} = S^\downarrow_{\text{TOA}} - S^\uparrow_{\text{TOA}}, \quad \epsilon = \frac{L_{\text{TOA}}}{L_{\text{s}}}, \quad \nabla_\phi F = S_{\text{net}} - L_{\text{TOA}},$$

where subscript TOA refers to the top of atmosphere and subscript s to the surface. Due to the assumption of the outgoing thermal radiation being proportional to $T^4$ and the approximation in the long-term average, we introduce a systematic error into the computation of the surface temperature. However, Voigt (2010) argues that the error is acceptably small for the typical case that $\epsilon \approx 1$ and for the usual surface temperatures. Comparing the surface temperatures from the energy-balance diagnostics with the model output, we find only small differences with largest values around the poles, exemplarily shown in Figure A2.

The surface-temperature difference between some equilibrium states A and B due to changes in net solar flux—comprising a differing solar constant, distribution of insolation, and albedo—can then be computed as

$$\Delta_{\text{B}-\text{A}}\langle T\rangle_{S_{\text{net}}} = \left[\frac{1}{\sigma\langle\epsilon\rangle_{\text{A}}}\left(\langle S_{\text{net}}\rangle_{\text{B}} - \langle\nabla_\phi F\rangle_{\text{A}}\right)\right]^{1/4} - \langle T\rangle_{\text{A}},$$

those due to changes in heat-flux divergence as

$$\Delta_{\text{B}-\text{A}}\langle T\rangle_{\nabla_\phi F} = \left[\frac{1}{\sigma\langle\epsilon\rangle_{\text{A}}}\left(\langle S_{\text{net}}\rangle_{\text{A}} - \langle\nabla_\phi F\rangle_{\text{B}}\right)\right]^{1/4} - \langle T\rangle_{\text{A}},$$

and those due to changes in effective emissivity as

$$\Delta_{\text{B}-\text{A}}\langle T\rangle_\epsilon = \left[\frac{1}{\sigma\langle\epsilon\rangle_{\text{B}}}\left(\langle S_{\text{net}}\rangle_{\text{A}} - \langle\nabla_\phi F\rangle_{\text{A}}\right)\right]^{1/4} - \langle T\rangle_{\text{A}}.$$



For the interpretation of these different contributions, it is helpful to realise that changes in the boundary conditions may well reduce or increase the net shortwave or longwave radiation globally, leading to global-mean cooling or warming. In contrast, by construction of the balance equation (1), the heat-flux divergence vanishes in the global average (e.g. North and Kim, 2017, their Section 5) and can thereby only affect the global-mean temperature via local changes and subsequent feedback effects affecting the shortwave or longwave radiation.

## 3 Sensitivity to continental configuration

### 3.1 Surface temperatures

Under Sturtian boundary conditions, the surface-air temperature is distinctly dependent on the continental configuration (Figure 4) with globally and annually averaged values ranging from $-8.9\,°C$ in the ideal dispersed configuration to $3.9\,°C$ in the southern-hemispheric (SH) polar-supercontinent configuration at an atmospheric $CO_2$ concentration of 250 ppm. The M+21 and L+13 reconstructions adopt intermediate climate states with values of $-1.7\,°C$ and $0.3\,°C$, respectively.

Simulations applying boundary conditions representative of the later Marinoan event, now at a $CO_2$ concentration of 200 ppm, can attest to these observations. The SH polar supercontinent facilitates a greater global-mean surface-air temperature of $3.6\,°C$, whilst the dispersed configuration presents the antithesis with a colder value of $-9.0\,°C$. The reconstructions once again adopt intermediate climate states with temperatures of $-6.7\,°C$ and $-1.8\,°C$ for the M+21 and L+13 palaeogeographies, respectively. The greater discrepancy between the reconstructions of the Marinoan as opposed to the Sturtian is potentially a result of the greater degree of dispersion present in the M+21 palaeogeography in the Marinoan glacial event due to the continued separation of the supercontinent Rodinia (Hoffman et al., 2017; Liu et al., 2013).

As noted before, the global land fraction slightly differs between the continental boundary conditions, which needs to be ruled out as a governing effect before discussing the individual energy-balance contributions. A small number of comparison simulations at 720 Ma with a manually-adjusted land fraction of ~17 % for all configurations exhibit effects on the surface-air temperature (the slight increase in continental area in the dispersed configuration even yields a Snowball at 250 ppm) but leave the order of temperatures and critical $CO_2$ concentrations among the boundary conditions unchanged.

### 3.1.1 Dispersed versus L+13 configurations

Let us now look at the individual energy-balance components (net solar radiation, meridional heat flux, and longwave emission) in order to understand the large difference in surface temperature between the dispersed configuration and the Sturtian L+13 reconstruction (Figure 4Ai–Av). The former case is significantly colder due to a great influence of changed net solar radiation, i.e. albedo, and a minor effect from increased global emissivity; see Figure A3 for the global contributions. Changes in the meridional heat flux mostly act to compensate for zonal changes in the surface temperature caused by the other energy-balance components. On global average, the meridional heat flux has virtually no effect on the surface temperature (Figure A3), as anticipated in Section 2.3. We proceed with discussing the components in more detail.







**Figure 4.** Absolute difference in surface temperature $\Delta T$ (Ai–Ci), land fraction $\Delta f_{\mathrm{land}}$ (Aii–Cii), land-snow and sea-ice fraction $\Delta f_{\mathrm{cryo}}$ (Aiii–Ciii), cloud fraction $\Delta f_{\mathrm{cloud}}$ (Aiv–Civ), and relative difference in column water vapour $\delta f_{\mathrm{vap}} = \Delta \overline{\langle f_{\mathrm{vap}} \rangle} / \overline{\langle f_{\mathrm{vap}} \rangle}_{\mathrm{ref}}$ (Av–Cv) between long-term-averaged attractor states upon changing the continental configuration for the Sturtian onset. Shown in Ai–Ci are the total temperature difference and differences due to changes in net solar radiation ('shortwave'), emissivity ('longwave'), and meridional heat-flux divergence ('meridional flux'). Differences are computed with reference to the L+13 720-Ma reconstruction (subtrahend).

Since neither the solar constant nor the distribution of insolation is changed between the compared setups, differences in *net solar radiation* only represent albedo effects. These effects can already be detected in the spinup of the simulations (not shown): after initialisation with nearly identical climatic conditions, the higher albedo and lower thermal inertia of high-latitude continents in the dispersed configuration lead to a strong cooling effect amplified by snowfall over land. As the simulations progress, the climate of the dispersed configuration consistently remains cooler than that of the L+13 reconstruction. Finally, upon ice margins encroaching into the mid-latitudes, the slightly larger continental area in the tropics further supports the cooler conditions in the dispersed case, favouring snowfall even there. In equilibrium, the most drastic differences in surface temperature occur over the sea-ice margins (Figure 4Ai). The cooling peaks around 40° S/N in the shortwave contribution are





correlated with the differences in cryosphere fractions (Figure 4Aiii) and likely the results of ice–albedo feedbacks at the ice margins. At the poles, the shortwave contribution yields higher temperatures in the dispersed configuration where the presence of continents prevents the formation of sea ice and snow does not sufficiently fall to compensate the change in albedo.

Changing the continental configuration can cause changes in the ocean circulation responsible for the *meridional heat transport* from lower to higher latitudes in the oceans (e.g. Jenkins, 1999), as seen also by modern-day analogies such as the role of the North Atlantic Current on the ocean temperatures of western Europe (Palter, 2015). Such changes can result in regional temperature effects which are difficult to detect in the zonal averages as in Figures 4. For the dispersed configuration, the ocean-surface currents are both diverted from the equator and slowed down as compared to the open ocean in the supercontinent configuration (shown in Figures 5A and C for the Marinoan conditions before Snowball inception). The situation is similar for the heat transport between the equator and the ice margins in the L+13 configuration (not shown).

Differences in *emissivity*, i.e. the greenhouse effect, within the modelled climate appear to have a localised influence dependent on surface temperature and primarily due to the water-vapour feedback: the lower temperature in the dispersed configuration is causative of the increased rates of evaporation and atmospheric concentrations of water vapour, again yielding lower temperatures. This demonstrates that emissivity is affected by the continental distribution indirectly and the result of a feedback rather than an initial cause of cooling/warming. The influence of emissivity on the global climate can be investigated further by distinguishing the localised influence of cloud fraction and water vapour. A comparison of Figures 4Ai, Aiv, and Av demonstrates that the longwave emission displays a great dependency on water-vapour concentration, modulated by cloud cover. Over the poles, the reduced cloud cover contributes to a cooling from reduced emissivity, which exceeds the warming from reduced ice cover.

A comparison of the dispersed configuration with the Marinoan L+13 reconstruction shows a very similar overall picture (Figures A4 and A5). The general tendency for cooling in a dispersed configuration is corroborated by Hyde et al. (2000), who adopted boundary conditions including a SH supercontinent originally derived by Dalziel (1997). By increasing the degree of dispersion between the continents, Hyde et al. (2000) note that the threshold in atmospheric $CO_2$ concentration for a Snowball inception is increased.

### 3.1.2 M+21 versus L+13 configurations

The Sturtian L+13 and M+21 reconstructions differ by 2 °C regarding their global-mean surface temperatures, with the M+21 configuration being the colder one. This difference is mainly caused by SH albedo differences and amplified by emissivity changes (Figures 4Bi–Bv and A3).

The largest difference in zonally averaged land fraction between the two configurations is found in middle and higher latitudes of the southern hemisphere, where we also observe a pronounced increase in snow- and sea-ice cover. The cooler southern hemisphere is therefore likely a result of the increased continental area.

For the Marinoan simulations the overall picture is very similar, but surface temperatures in the L+13 and M+21 reconstructions differ more strongly, mainly due to albedo differences (Figure A5). Considering zonal averages (Figure A4Bi–Bv), the Marinoan configurations come with locally larger absolute differences in continental area and the cryosphere fractions than





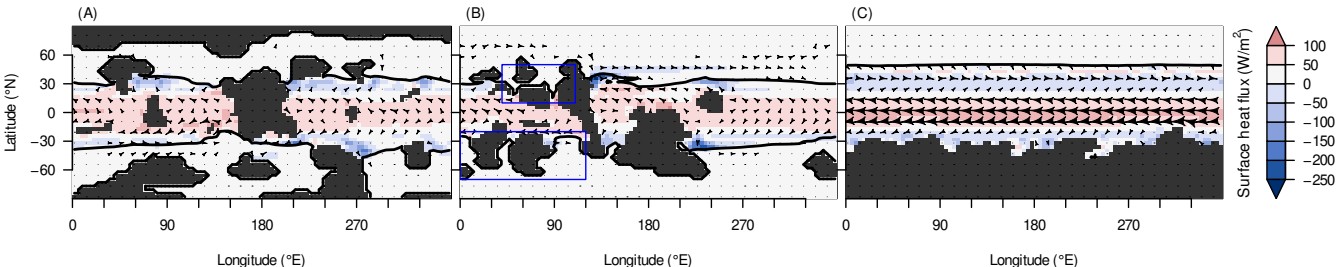

**Figure 5.** Long-term-mean sea-ice cover (50-% contours as thick black lines), surface ocean velocity (arrows with lengths proportional to the velocity), and heat flux at the ocean surface (colours, positive values indicate the ocean gaining heat) for Marinoan (650 Ma) equilibrium simulations using the dispersed (A), M+21 (B), and polar-supercontinent configuration (C). $CO_2$ concentrations are 200 ppm (A), 160 ppm (B), and 20 ppm (C). Blue rectangles in panel B indicate ocean regions featuring local sea-ice protrusions.

415 the Sturtian equivalents. Additionally, the slightly greater area of low- and middle latitude landmass present in the Marinoan M+21 relative to the L+13 reconstruction reduces the absorption of solar radiation between the sea-ice margins.

For both the Sturtian and the Marinoan, emissivity and meridional heat flux reflect the zonal cooling or warming caused by albedo differences. Nevertheless, the Marinoan M+21 configuration provides an example of local protrusions in the sea-ice front: due to the positioning of continental clusters, several middle-latitude ocean parts between 0 and 120° E are secluded from 420 ocean currents and, without an influx of warmer equatorial water, adopt a cooler regional climate (blue rectangles in Figure 5).

### 3.1.3 Polar supercontinent versus L+13 configurations

The SH polar supercontinent stands out as the warmest configuration (Figures 4 and A3). Again, the largest part of the temperature difference in comparison to the Sturtian L+13 configuration results from net shortwave differences (Figure A3Ci–Cv). Almost the entire supercontinental domain has surface temperatures of several degrees with the warming only due to albedo 425 changes. The differences in cryosphere fractions (Figure 4Ciii) imply that it is the absence of sea ice which increases the albedo despite the presence of snow on land.

Apart from the albedo, the difference in meridional heat transport seems to play an important part: the continental margin obstructs equatorial heat from bypassing ~30° S latitude via ocean currents. The resulting warming in this zone exceeds the cooling from snow cover. This enforces a decreased temperature reduction during the SH winter months, minimising sea- 430 ice advancement between the equator and 30° S due to the stored thermal energy, as illustrated in Figure 5C. Note that this configuration is not represented anywhere in geologic history and so this mechanism likely does not elucidate any climate phenomenon currently under study.

Emissivity changes only partly reflect a simple water-vapour feedback. Over the supercontinent, cloud cover is homogeneously lower than in the L+13 configuration and gives rise to increased longwave emission. This effect is compensated by 435 increased water-vapour concentrations only over the South Pole. Despite this cooling through longwave radiation, most of the southern hemisphere is warmer in the supercontinent configuration and can partly explain the warmer conditions in the north-





ern hemisphere through an ice–albedo feedback around the sea-ice margin. Perhaps another reason for a warmer NH climate is the absence of snow on land.

Very similar observations can be made under Marinoan conditions (Figures A4 and A5).

## 3.2 Equilibrium thresholds

The susceptibility of the planet under each continental configuration to a global glaciation is reflected also in the critical $CO_2$ thresholds, i.e. the highest atmospheric $CO_2$ concentration at which a Snowball inception occurs, shown together with the coldest non-Snowball states in Figure 6. There exists a correlation between these thresholds and the surface-air temperature for given $CO_2$ concentrations. This result is plausible because a colder climate state typically features a larger sea-ice extent—and should therefore be closer to some glaciation threshold—than a warmer state, given that both configurations share the same critical sea-ice extent necessary for a Snowball inception. Contradicting this expectation however, we find the M+21 to be more resistant to global freezing than the warmer L+13 reconstructions at both the Sturtian and the Marinoan onset. The cause is a stable sea-ice extent with margins around the Hadley-cell boundaries ('Hadley state', Feulner et al., 2023), which is only supported in the M+21 configuration and unstable in the L+13 case (insets in Figure 6).

Under Sturtian boundary conditions, the equilibrium thresholds range from 240–250 ppm of atmospheric $CO_2$ in the dispersed configuration to 30–40 ppm in the more resistant supercontinent geography. The L+13 and M+21 palaeogeographies adopt intermediate values of 130–140 ppm and 90–100 ppm, respectively.

Moreover, under the boundary conditions synonymous with the Marinoan glaciation, critical $CO_2$ thresholds are lower than those observed for the Sturtian event. This includes a threshold range of 10–20 ppm for the resistant supercontinent configuration and 190–200 ppm for the more susceptible dispersed configuration. The nonidealised palaeogeographic reconstructions again support intermediate ranges of 170–180 ppm and 160–170 ppm of atmospheric $CO_2$ for the L+13 and M+21 configurations, respectively.

The increased resistance to glacial advance in the Marinoan simulations, as indicated by the colder climate states required, can be attributed to the higher solar irradiance in concurrence with the Bahcall et al. (2001) model. The increased insolation incident on Earth's surface results in a global net increase in surface temperatures (Feulner and Kienert, 2014), which requires a lower greenhouse influence to negate before the albedo feedback can facilitate global ice coverage.

## 4 Sensitivity to orbital geometry

### 4.1 Surface temperatures

A comprehensive comparison of long-term- and global-mean surface-air temperatures for 51 different combinations of the orbital parameters obliquity $\varepsilon$, eccentricity $e$, and argument of perihelion $\omega$ (Figure 7) shows variations between $-5.4\,°C$ ($\varepsilon = 24.5°$, $e = 0$) and $-2.1\,°C$ ($\varepsilon = 22°$, $e = 0.069$, $\omega = 90°$) with the Sturtian L+13 palaeogeography and an atmospheric





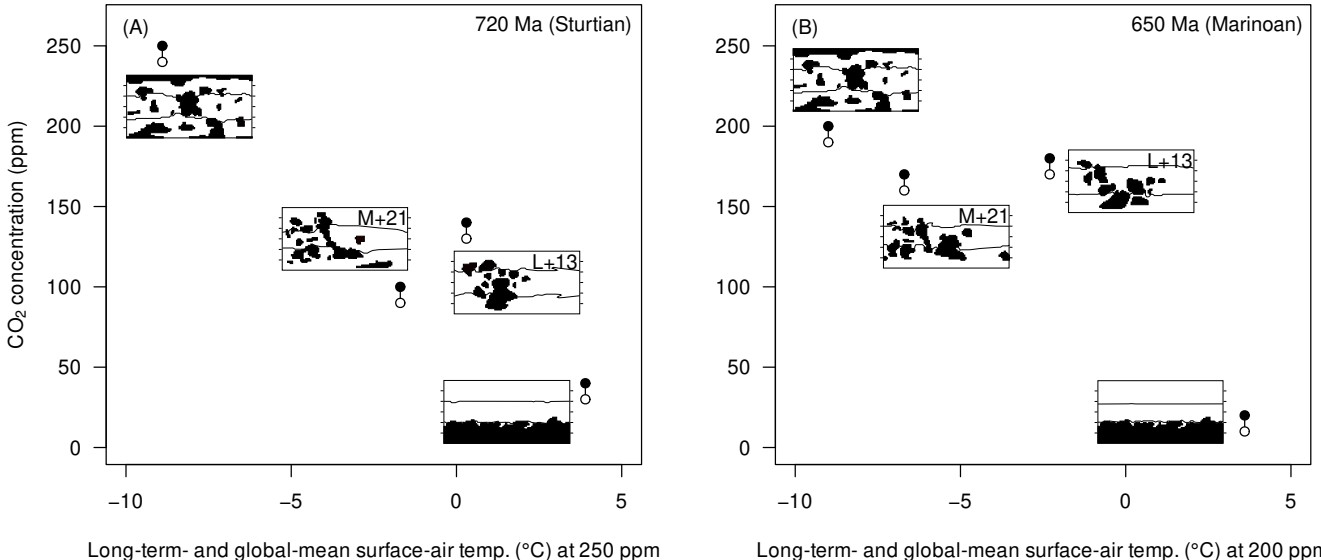

**Figure 6.** Long-term- and global-mean surface-air temperature at fixed $CO_2$ levels and $CO_2$ concentration just above (filled circles) and below (open circles) the Snowball-inception threshold for different Sturtian (A) and Marinoan boundary conditions (B). Surface-air temperatures are shown for a $CO_2$ concentration of 250 ppm (A) and 200 ppm (B). Insets show the continental configurations used in the simulations and the sea-ice margins (sea-ice concentration of 0.5) in the filled-circle cases, i.e. just above the threshold.

$CO_2$ concentration of 150 ppm as boundary conditions. For all values, see Table B1. We emphasise that this comparison aims at orbital effects in very cold climate states; sea-ice margins extend to around 30–40° S/N.

One can observe distinct dependencies of surface-air temperature on $\varepsilon$ and $e$: First, increasing $\varepsilon = 22° \rightarrow 23.5°$ while leaving $e$ and $\omega$ fixed reduces temperature by 0.11–0.81 °C in all 17 cases; in 14 out of 17 cases, the increase $\varepsilon = 23.5° \rightarrow 24.5°$ leads to further cooling of up to 0.46 °C (and warming of up to 0.61 °C in three cases). Second, increasing $e$ as $0 \rightarrow 0.03$ for fixed $\varepsilon$ and $\omega$ yields a surface-air warming of up to 0.86 °C in 23 out of 24 cases (and cooling of 0.20 °C in one case); in all cases, the increase $e = 0.03 \rightarrow 0.069$ accounts for a global surface-air warming of 1.1–2.4 °C. Regarding the complete parameter ranges, the $22° \rightarrow 24.5°$ increase of $\varepsilon$ causes a global cooling of 0.21–1.0 °C, while the $0 \rightarrow 0.069$ increase of $e$ is accompanied by a

warming of 1.3–2.7 °C.

The isolated effects of orbital parameters on surface-air temperatures are illustrated by means of a few examples in Figure A6. In addition to the essentially cooling and warming effects of increased $\varepsilon$ and $e$, respectively (left and middle panels of the figure), the isolated effect of $\omega$ is less clear. For the largest eccentricity, $e = 0.069$, one cycle of $\omega$ covers two temperature peaks at $\omega$ near solstices (December and June) and two minima at $\omega$ near equinoxes (September and March). This behaviour

holds for all three cases with $e = 0.069$, but the responses to variations of $\omega$ with $e = 0.03$ are inconsistent; we neglect the latter in the detailed discussions of $\omega$-dependence below.




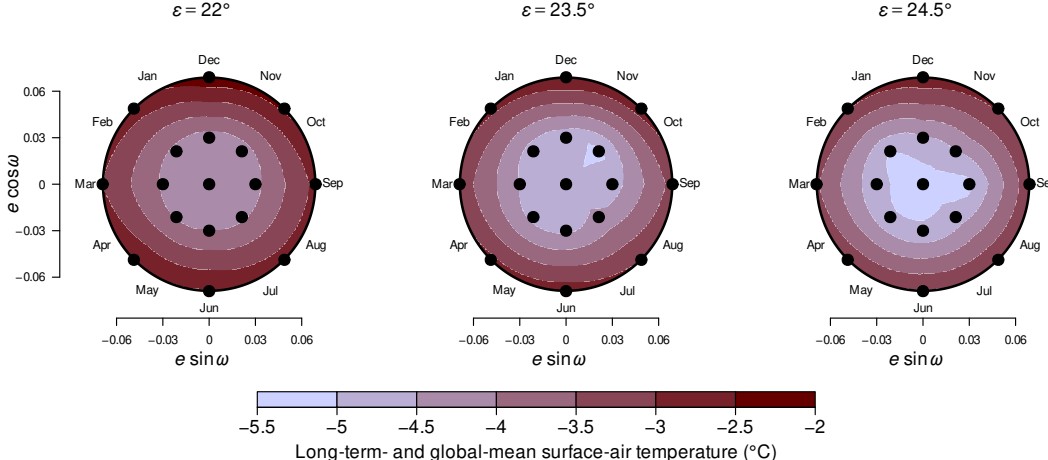

**Figure 7.** Mean surface-air temperatures for different combinations of orbital parameters. Temperatures are taken from the warmest attractors supported for a $CO_2$ concentration of 150 ppm and the Sturtian L+13 palaeogeography. For each obliquity $\varepsilon$, the eccentricity $e$ and argument of perihelion $\omega$ are drawn as radial and angular coordinates, respectively. Abbreviations of months clarify the time of perihelion passage. Black dots denote geometries for which model simulations exist. Values between the dots result from linear interpolation.

### 4.1.1 Obliquity

As an example, we consider the zonal differences in the surface energy balance due to changing the orbital geometry from $\{\varepsilon = 22°,\ e = 0\}$ to $\{\varepsilon = 23.5°,\ e = 0\}$ (Figure 8Ai–Av). For example, a negative temperature difference in this diagram
implies a cooling contribution upon increasing the obliquity $\varepsilon = 22° \rightarrow 23.5°$.

Since our obliquities are well below $55°$, the increase in $\varepsilon$ generally yields a more evenly distributed annual-mean incoming solar flux at the top of atmosphere (e.g. Linsenmeier et al., 2015, their Figure 2), as also seen in our example (Figure 8Aii). This can explain why the change in the net solar flux causes a surface cooling around the equator and a surface warming around the poles. The remaining features of the shortwave contribution are correlated with the differences in cryosphere fractions
(Figure 8Aiii) and seem to illustrate, at least, processes of local ice–albedo feedback. This feedback can also explain minor reductions seen in the sea-ice concentration and snow cover southwards of around $45°$ S. Nevertheless, the global temperature differences are dominated by shifts of the cryosphere margins, which lie within the region of reduced insolation.

Changes in emissivity or heat-flux divergence mainly cause smaller differences in the surface temperature than the net solar flux. As an exception, the heat-flux divergence contributes to a cooling on the poleward sides of the cryosphere margins,
exceeding the warming solar contribution. We understand this as resulting from the decreased meridional temperature gradient in these regions: the local cooling around the cryosphere margins reduces that gradient polewards and thereby reduces the poleward heat transport there.

The local emissivity changes generally correlate well with the local balance between shortwave and transport contributions (not shown separately, but evident from Figure 8Ai). We can attribute these changes to two positive feedbacks. First, a decrease





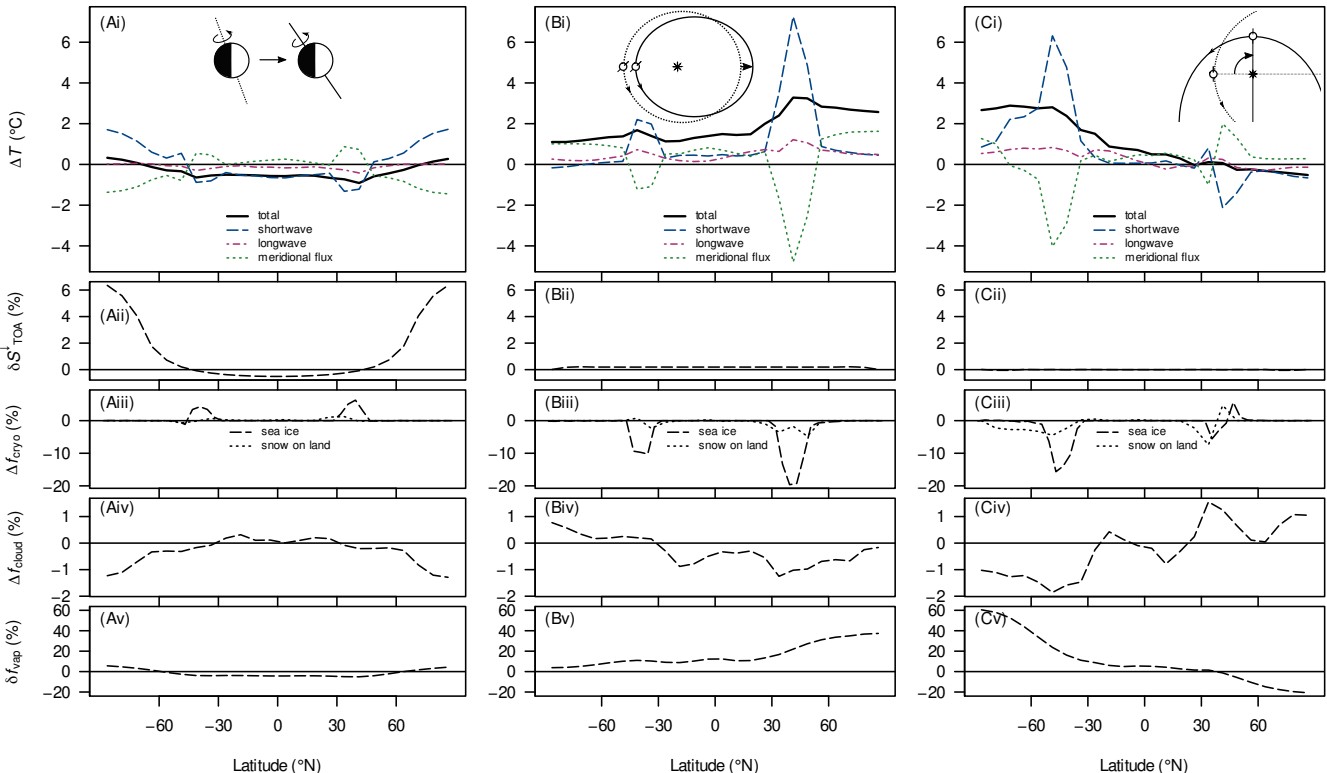

**Figure 8.** Absolute difference in surface temperature $\Delta T$ (Ai–Ci), relative difference in top-of-atmosphere incoming solar flux $\delta S_{\mathrm{TOA}}^{\downarrow} = \Delta \langle \overline{S_{\mathrm{TOA}}^{\downarrow}} \rangle / \langle \overline{S_{\mathrm{TOA}}^{\downarrow}} \rangle_{\mathrm{ref}}$ (Aii–Cii), absolute differences in land-snow and sea-ice fraction $\Delta f_{\mathrm{cryo}}$ (Aiii–Ciii) and the cloud fraction $\Delta f_{\mathrm{cloud}}$ (Aiv–Civ), and the relative difference in column water vapour (Av–Cv) between long-term-averaged attractor states upon changing individual orbital parameters. Shown in Ai–Ci are the total temperature difference and differences due to changes in net solar radiation ('shortwave'), emissivity ('longwave'), and meridional heat-flux divergence ('meridional flux'). Ai–Av show effects of an increased obliquity ($\varepsilon = 23.5°$ as compared to the reference $22°$) with $e = 0$ fixed; Bi–Bv those of an increased eccentricity ($e = 0.069$ versus the reference 0.03) with $\varepsilon = 23.5°, \omega = 225°$ fixed; and Ci–Cv those of a shifted argument of perihelion ($\omega = 90°$ versus the reference $180°$) with $\varepsilon = 22°, e = 0.069$ fixed. All differences are computed for zonal-mean magnitudes. Remaining boundary conditions are $S_0 = 1283\,\mathrm{W\,m^{-2}}$, a $CO_2$ concentration of 150 ppm, and the Sturtian L+13 palaeogeography. In the inset schematics, circles depict Earth and asterisks the Sun. New configurations (minuend) are drawn with solid, reference configurations (subtrahend) with dotted lines. Schematics are not to scale.

in surface temperature lowers the atmospheric water-vapour concentration and thus raises the effective emissivity; this well-known water-vapour feedback occurs mainly in the tropics and mid-latitudes, but also acts to enhance the high-latitude warming (Figure 8Av). Second, the cooling around the cryosphere margins comes with a local decline of stratus clouds, also increasing the effective emissivity (Figure 8Aiv).

It is noteworthy that we observed a cooling with increased obliquity, while many studies report the opposite effect (e.g. Mantsis et al., 2011; Nowajewski et al., 2018; Brugger et al., 2019; Landwehrs et al., 2021). While these studies only investi-





gated comparatively warm climate states with ice margins above $55°$, the cooling effect shown here is consistent with the very cold late-Carboniferous/early-Permian simulations of Feulner (2017). In fact, for climate states with an ice line above $55°$, our model simulates mostly rising temperatures with increased obliquity (Figure A8).

All the discussed features are present in almost all other comparisons with differing obliquity but fixed other parameters
(not shown). There exist two exceptions, where increasing the obliquity from $23.5°$ to $24.5°$ raises the temperature (e.g. see Figure A6), but they are not discussed in detail here.

### 4.1.2 Eccentricity

We investigate the effects of increased eccentricity by comparing the orbital geometries $\{\varepsilon = 23.5°, e = 0.03, \omega = 225°\}$ and $\{\varepsilon = 23.5°, e = 0.069, \omega = 225°\}$ (Figure 8Bi–Bv), corresponding to perihelion in early May. In this example, we find globally
increased surface temperatures as well as a hemispheric asymmetry regarding the total temperature differences.

Changes in the net solar flux contribute almost globally to the warming. A background contribution of well below $+1\,°C$ can be explained by the globally increased top-of-atmosphere insolation (Figure 8Bii), which is due to the smaller semi-minor axis of the higher-eccentricity orbit resulting in a reduced mean Sun–Earth distance. Subannually however, this distance oscillates and the increased eccentricity yields both a more pronounced perihelion and aphelion. For most values of $\omega$, as the one shown
here, this increases the already existing net insolation difference between the hemispheres. This difference induces accelerated ice melting and strong warming during spring and summer in one hemisphere, and delayed melting and weak relative cooling in the other. In principle, we see the same effect for varying $\omega$, which will be discussed below.

Outside the cryosphere margins, the warming contributions from the meridional heat-flux divergence are larger than those from the net solar flux. The local warming around the ice margins affects the meridional temperature gradients, the latter now being reduced around the equator and enhanced over the polar ice caps. In the model, the correspondingly changing meridional
heat transport causes the warming in both regions.

Changes in emissivity largely reflect the positive feedbacks already seen for increasing the obliquity (Figure 8Biv–Bv).

Depending on $\omega$, the hemispherically asymmetric warming upon increasing the eccentricity as $0.03 \rightarrow 0.069$ can vary significantly, as discussed below. The increased global-mean insolation, however, is seen in all cases (not shown).

### 530 4.1.3 Argument of perihelion

For changes in the argument of perihelion, we look at the two orbital geometries $\{\varepsilon = 22°, e = 0.069, \omega = 180°\}$ and $\{\varepsilon = 22°, e = 0.069, \omega = 90°\}$ (Figure 8Ci–Cv), with the choices of $\omega$ representing the largest difference in surface-air temperature for given $\varepsilon$ and $e$ (cf. Figure A6).

Shifting $\omega$ does neither change the meridional distribution nor the global amount of incoming solar flux in the annual
average (Figure 8Cii). However, it affects the seasonal differences in insolation. At $\omega = 180°$, perihelion and aphelion occur in March and September, respectively, i.e. at times with intermediate insolation in both hemispheres. In comparison, at $\omega = 90°$—perihelion in December, aphelion in June—the seasonality of the solar flux is amplified in the southern and dampened in the northern hemisphere. Thus, during the ice-melting seasons (spring through summer), the configuration $\omega = 90°$ lowers the





incoming solar flux in the northern and raises it in the southern hemisphere, which affects the ice-melting rates and contributes
to the observed hemispheric warming/cooling asymmetry. Since the absolute increase in solar flux is larger in the southern
hemisphere than the absolute decrease in the northern hemisphere, the SH warming turns out stronger than the NH cooling.
This is the main reason for the overall warming upon shifting $\omega = 180° \rightarrow 90°$.

Shortwave reflection from snow in southern middle and high latitudes, particularly over the largest continent, is reduced
in the annual average at $\omega = 90°$ (Figure 8Ciii). This contributes to a significant warming in these latitudes and is a plausible
reason for the configurations $\omega = 90°$ being slightly warmer than $\omega = 270°$. In order to test this hypothesis, we ran a number of
simulations at varying $\omega$ with the continents mirrored at the equator. A comparison with the original results (Figure A9) shows
that the difference between the temperature maxima at $\omega = 90°$ and $270°$ can be explained by the continental configuration
and, in particular, the location of the largest continent.

Changes in the meridional heat flux counteract the shortwave contributions around the ice margins and around the North
Pole, but play a significant role in warming the tropics and the South Pole.

Emissivity differences yield temperature changes of smaller magnitude and the same sign as the total temperature difference,
again reflecting the positive water-vapour and stratus-cloud feedbacks.

Comparing the configurations with $\omega = 180°$ and $270°$, the results look very similar except for the strong warming then
occurring in the northern hemisphere.

## 555 4.2 Equilibrium thresholds

Upon lowering the $CO_2$ concentration to the critical levels, the sea-ice margins undergo a sudden shift from around 40° S/N
('Ferrel state', Feulner et al., 2023) to the equator for 8 of the 9 tested orbital geometries (Figure A10). Only the configuration
$\{\varepsilon = 22°, e = 0\}$ supports stable ice margins at around 30° S/N, i.e. the poleward Hadley-cell boundaries ('Hadley state',
Feulner et al., 2023), before transitioning into a Snowball state. The threshold for Snowball inception, here defined as the
highest $CO_2$ concentration at which the transition occurs, lies between 108 ppm for $\{\varepsilon = 22°, e = 0.069, \omega = 270°\}$ and
140 ppm for $\{\varepsilon = 24.5°, e = 0\}$.

A comparison of the $CO_2$ thresholds with the surface-air temperatures at 150 ppm shows that lower temperatures favour
Snowball inceptions at higher $CO_2$ concentrations (Figure 9). One interesting exception from this rule seems to be that ge-
ometries with $\omega = 90°$—perihelion in December—are more susceptible to full glaciation than with $\omega = 270°$—perihelion in
June—for the same remaining boundary conditions, although the latter are generally colder than the former at 150 ppm. In fact,
at $CO_2$ concentrations closely above the thresholds, the order is reversed and $\omega = 90°$ yields cooler surface-air temperatures
than $\omega = 270°$. Through an exemplary comparison of energy-balance contributions (Figure A11) we identify changes in the
sea-ice cover as the main reason. Well above the glaciation threshold (115 ppm and more), shifting the perihelion from June to
December leaves the global sea-ice cover essentially unchanged, as hemispheric differences balance out; differences in snow
cover dominate the global temperature differences. Closer to the threshold (112 ppm), however, this shift in perihelion allows
SH sea ice to form in the centre of a large subtropical gyre despite the overall SH decrease of sea-ice concentration.

The temperatures and thresholds presented above are the results of unperturbed simulations, taking thousands of years to reach equilibrium-like states. In reality, the orbital geometry varies on similar timescales and thus forces the climate into a permanent nonequilibrium state. Our results should therefore be understood as providing lowers and upper bounds for the actual effects.

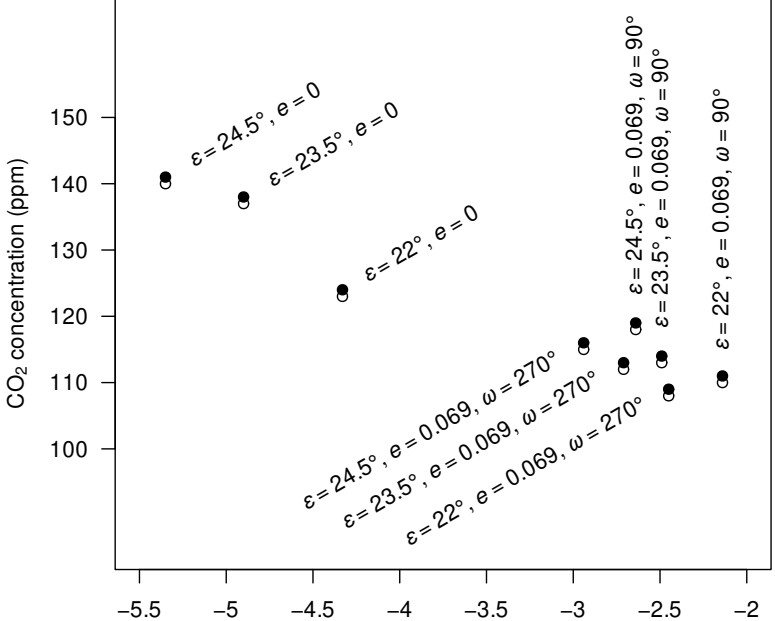

**Figure 9.** Atmospheric $CO_2$ concentrations just above (filled circles) and below (open circles) Snowball inceptions under unperturbed conditions and the corresponding long-term- and global-mean surface-air temperatures at 150 ppm, shown for nine different orbital geometries as indicated by the text labels.

## 5 Sensitivity to volcanism

### 5.1 Single global volcanic perturbations

The shifts in $CO_2$ thresholds, below which the climate enters a Snowball attractor, due to single global volcanic perturbations are shown in Figure 10. Besides the trivial observation that these shifts tend to increase for stronger perturbations, we see significantly different results depending on the orbital geometry. For example, a monotone and somewhat linear increase is seen for the configurations $\{\varepsilon = 22°, e = 0\}$ and $\{\varepsilon = 24.5°, e = 0.069, \omega = 270°\}$. Initialising the simulations on the coldest partially ice-covered attractors, the strongest nominal volcanic peak forcing of $-40\,\mathrm{W\,m^{-2}}$ yields $CO_2$-threshold shifts of



up to 31 or 49 ppm, respectively. However, strong perturbations initalised on the warmest partially ice-covered attractor of the former orbital configuration are much less effective with a maximum shift of 14 ppm, while in the latter configuration the initial

attractor plays almost no role. Strikingly, even the most extreme perturbations are barely able to trigger a Snowball inception in the configuration $\{\varepsilon = 23.5°, e = 0\}$. Let us now briefly investigate why, first, the initial state can make such a large difference in one orbital geometry ($\varepsilon = 22°$) and, second, another orbital geometry ($\varepsilon = 23.5°$) can pose such a strong resilience against perturbations.

In the theory of dynamical systems, a perturbation can only shift a system to a new attractor if the system is forced to cross a

state separating the two basins of attraction. Consequently, differences in the response to single perturbations should be related to differences in the topology of the attractors. In Figure 11, we plot an approximate picture permitting conclusions about the attractors and their basins of attraction in the three different orbital configurations. Attractors are sampled by comprehensive sets of unperturbed hysteresis experiments between 175 or 150 ppm and the respective Snowball thresholds, projected onto the phase space spanned by the $CO_2$ concentration and the global-mean surface-air temperature. The temperatures are obtained

by the same procedure as for finding the thresholds for different orbital geometries (described in Section 2.2.2), but attractors coexisting with the warmest branches are reconstructed by incrementally increasing the $CO_2$ concentration starting from selected states. This procedure is similar to the reconstruction of stable branches described in Brunetti and Ragon (2023) (their Method II), but an important difference is that we may exclude 'hidden' attractors which are not accessible via mere changes in the boundary conditions. However, since we always allow 1000 yr or more for model relaxation, the reconstructed attractors

should be very precise, compared to other methods (Brunetti and Ragon, 2023).

The temperatures shown in the figure result from the annual means, averaged over the last 200–1000 yr of the respective simulations. Selected perturbations are shown together with the respective long-term model response in order to delineate the basins of attraction. However, it should be noted that the model typically relaxes over several hundreds of years and the surface temperature can show an 'overshooting' behaviour, when further cooling after the peak forcing and subsequent relaxation to a

warmer state is observed. The basins of attraction can therefore not be inferred from the peak forcing alone.

As previously mentioned for the ice-line latitudes (Figure A10), the configuration $\{\varepsilon = 22°, e = 0\}$ supports an intermediate cold attractor, which—judging from the hysteresis experiments—seems to be absent for the other two orbital geometries. The intermediate attractor consists of 'Hadley states' (Feulner et al., 2023), i.e. sea-ice margins lying at the poleward Hadley-cell boundaries, whereas the common stable states have ice margins around $40°$ S/N and are referred to as 'Ferrel states'. This

dichotomy explains the large dependence of the results on the initial conditions in the former orbital geometry. We observe that some volcanic eruptions in the configuration $\{\varepsilon = 23.5°, e = 0\}$ force the model into similar, stable Hadley states, preventing the model from a Snowball inception and thereby rendering this configuration so resilient against perturbations. However, these intermediate stable states do not seem to be accessible by the quasi-static hysteresis experiments.

### 5.2 Single zonal volcanic perturbations

For the orbital geometry $\{\varepsilon = 22°, e = 0\}$, Figure 12 shows that the zonal perturbations are often comparable to the global perturbations regarding their effect on the $CO_2$ threshold, but this effect can vary with the latitude of the eruption. The latitude




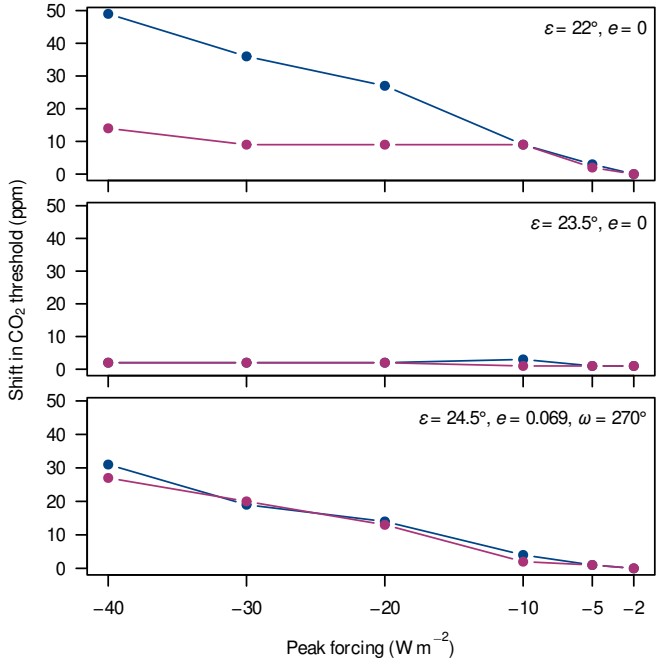

**Figure 10.** Shift in $CO_2$ thresholds due to single global volcanic perturbations in three different orbital configurations, each indicated within the panels. Perturbations were initialised either on the warmest (red) or coldest partially ice-covered attractor (blue) simulated at the respective $CO_2$ level. For example, a shift of 10 ppm means that a perturbation triggers a Snowball inception at a $CO_2$ concentration 10 ppm above the one expected from unperturbed simulations.

dependence tends to be stronger for the larger perturbations and is especially pronounced for eruptions with a nominal peak forcing of $-10\,\mathrm{W\,m^{-2}}$ or beyond, when perturbing equilibrium states on the coldest partially ice-covered attractor. At these magnitudes, eruptions have the largest effect, and mostly exceed the effects of comparable global perturbations, if occurring
close to the equator. For example, an eruption with a nominal peak forcing of $-20\,\mathrm{W\,m^{-2}}$, initialised on the colder attractor, causes a shift of 31 ppm when centred at $15\,°$N and 6 ppm when centred at $58\,°$N; in comparison, the global perturbation shifts the threshold by 27 ppm. Note that there exist no simulations with peak forcings of $-30$ or $-40\,\mathrm{W\,m^{-2}}$ for eruptions at $58°$ S/N, since the corresponding AOD values would yield a locally negative insolation using the assumed proportionality between AOD and reduction of the solar constant.
The mechanism responsible for this latitude dependence becomes clearer by looking at the time-dependent response of the surface-air temperature following eruptions of the same magnitude but at different latitudes (Figure A12). In the shown example at 150 ppm, the eruption at $15°$ S triggers a Snowball inception, while the eruption at $58°$ S is followed by a relaxation towards a colder partially ice-covered attractor. The evolution of surface-air temperature following the tropical eruption at $15°$ S is very similar to the response following the global perturbation (Figure A12, middle and top row, respectively): In both
cases we find a transient relaxation after the peak forcing but a longer-term cooling, eventually leading to a Snowball Earth.

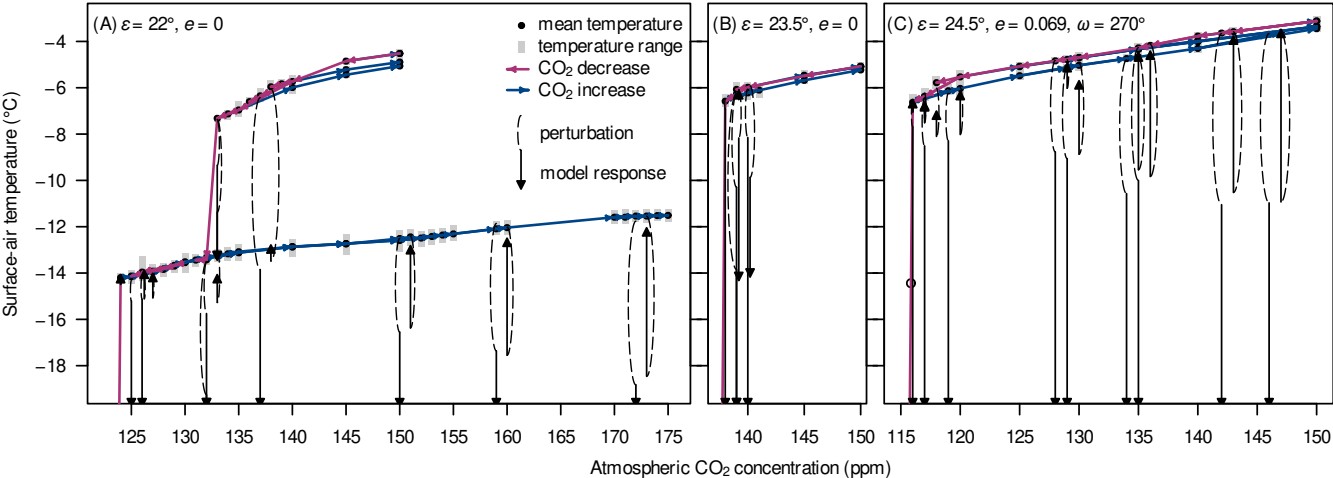

**Figure 11.** Long-term- and global-mean surface-air temperatures (filled circles) and temperature ranges (grey) at different $CO_2$ concentrations in hysteresis experiments (red and blue arrows), supplemented with selected global volcanic eruptions (dashed black curves) and the model response following the peak forcing (black arrows) for orbital geometries $\{\varepsilon = 22°, e = 0\}$ (A), $\{\varepsilon = 23.5°, e = 0\}$ (B), and $\{\varepsilon = 24.5°, e = 0.069, \omega = 270°\}$ (C). Arrows pointing beyond the lower panel boundaries indicate transitions to Snowball attractors with surface-air temperatures of around $-60\,°C$. The open circle in panel C indicates that the simulation was discontinued at this temperature due to numerical instability in the course of a Snowball inception.

More strikingly, both perturbations lead to rather homogeneous temperature reductions in the first years. The similarities come as no surprise since our tropical eruption scenarios feature a fast spreading of aerosols across all latitudes (Figure A1). In comparison, the eruptions in higher latitudes only cause hemispherical reductions of the solar flux. The seasonal evolution of surface-air temperature shows a pattern: During the first years with high aerosol concentrations, the cooling is stronger in

the respective summer hemisphere. However, for the simulations transitioning into a Snowball state, the longer-term cooling happens faster on the respective winter hemisphere. This is plausible since the initial cooling results from a direct reduction of incoming radiation being stronger in absolute value for higher levels of insolation, whereas the long-term cooling comes with the formation of ice being more effective during winter. For the eruptions in higher latitudes, aerosols become less dispersed over the globe and thus lead to a stronger hemispheric response (e.g. Figure A12, bottom row). Still, the cooling leaves the

tropical ocean ice-free and therefore cannot trigger a Snowball inception.

    The impact of the seasonal timing of an eruption cannot be thoroughly investigated with our simulations, but we observe that high-latitude events in the northern hemisphere often yield stronger temperature reductions (not shown) and threshold shifts (Figure A12) than in the southern hemisphere. Whether this is due to the eruptions starting in January or to continental asymmetry is difficult to determine here.





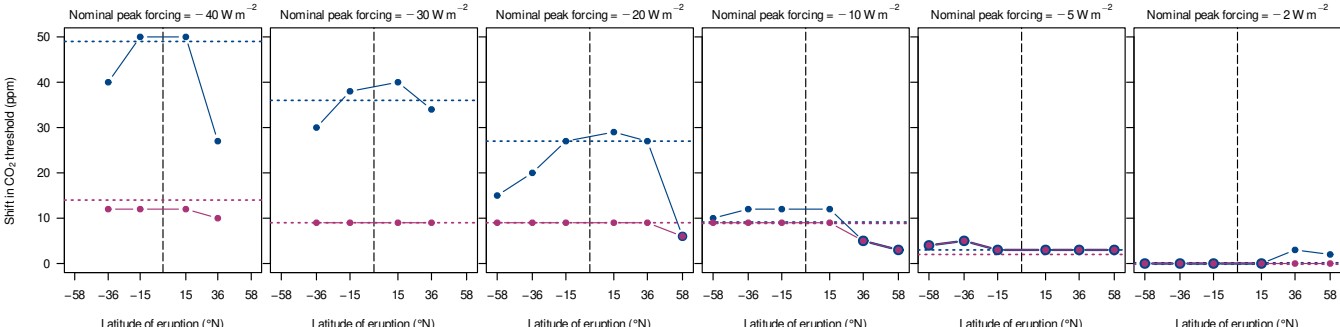

**Figure 12.** Shift in $CO_2$ thresholds due to single zonal volcanic perturbations in the orbital configuration $\{\varepsilon = 22°, e = 0\}$. Different panels contain the response to different magnitudes of the nominal peak forcing, as indicated above each panel. Perturbations were initialised either on the warmest (red) or coldest partially ice-covered attractor (blue) simulated at the respective $CO_2$ level. Dotted lines indicate shifts due to global perturbations of the same magnitude.

### 5.3 Sequences of global volcanic perturbations

The different scenarios of sequential volcanic eruptions with a globally-homogeneous forcing are able to shift the $CO_2$ threshold for a Snowball inception by a wide range between 17 and 52–56 ppm (Figure 13). The reference scenario with parameters $\{p = 0.1, \beta = 1, \xi = 0.3, u = -1.5 \, \text{W m}^{-2}\}$ triggers a tipping 27–31 ppm above the equilibrium threshold in the selected orbital configuration. The smallest shift is realised by a scenario with a lower shape parameter ($\beta = 0.5$), while the largest shift is observed for an increased eruption probability ($p = 0.3$).

The top-row panels of Figure 13 provide comparisons with the expected ranges from single global volcanic perturbations initialised on the coldest (blue rectangles) and the warmest (red) attractors coexisting at given $CO_2$ concentrations. For example, the expected range for the reference sequence, with the strongest peak forcing of $-22.8 \, \text{W m}^{-2}$, is determined by the shifts from single global perturbations with peak forcings of $-20$ and $-30 \, \text{W m}^{-2}$. Although the sequential perturbations are initialised on the warmest attractors, all scenarios lie well above the expected ranges shown in red. Instead, five out of six cases are within or above the ranges of single perturbations acting upon the coldest equilibrium states. The reason for this is that, as mentioned before, the sequences tend to render the warmer attractors unstable and nudge the model towards the coldest ice-covered states.

Let us now investigate the differing effects of the scenarios by looking at the strongest eruptions in the time series, marked by red arrows in Figure 13. Ignoring the last scenario ($p = 0.3$) for a moment, the impact strictly increases with the magnitude of the largest event, which is mostly affected by the shape parameter $\beta$ ($-12.1 \, \text{W m}^{-2}$ for $\beta = 0.5$; $-33.4 \, \text{W m}^{-2}$ for $\beta = 1.5$). We can distinguish three cases: first, the scenario with a threshold of $u = 0 \, \text{W m}^{-2}$ falls between the red and blue ranges. The reason is that in the respective simulation ending in a Snowball state, the largest eruption occurs while the climate is still on the warmest attractor (not shown); on the intermediate coldest attractor, the eruptions stay weaker than $-17.4 \, \text{W m}^{-2}$ and, in fact, produce a result well between single perturbations of $-10$ and $-20 \, \text{W m}^{-2}$ (shown by the blue rectangle for the scenario





$\beta = 0.5$). Second, three scenarios stay within the ranges from single perturbations. For these, the effect of a sequential forcing is, at least partly, dominated by the single largest eruption.

Third, two scenarios, although staying below the magnitude of certain single eruptions, have impacts that go beyond these single events. As this cannot be explained by the largest eruptions alone, the superposition of eruptions must play a role. Causing
permanent presence of aerosols from the frequent smaller eruptions, the superposition lowers the long-term temperature in almost all simulations. In the two cases of interest, the most frequent eruptions are stronger ($u = -3\,\mathrm{W\,m^{-2}}$, see Figure 3, B) or even more frequent ($p = 0.3$) as compared to the reference scenario, yielding an enhanced long-term cooling (not shown). This seems to amplify the sensitivity to large eruptions considerably.

It is worth noting that a recent study investigated noise-induced tipping towards Snowball states in an energy-balance model
under different noise types (Lucarini et al., 2022). For noise characterised by probability distributions with long tails, allowing for extreme events such as volcanic eruptions, the shape of the noise was found to be of much less importance for the tipping behaviour than single extreme events triggering the tipping. Our results, obtained by prescribing noise-like volcanism, provide evidence along similar lines but additionally highlight the role of a one-sided low-magnitude noise changing the long-term surface temperature. The major difference is the symmetry of noise: Our highly asymmetric variability in the temperature
signal—roughly symmetric internal variability overlaid by one-sided volcanism—stems from the focus on volcanism and ignores some other global-scale extreme events, including those resulting from internal climate dynamics. On the other hand, it might be interesting to systematically investigate the effect of asymmetric noise in simple models.

Although the tested scenarios are based on the analysis of historical data by Ammann and Naveau (2010), it is unclear whether they are representative of the volcanic modes during the Cryogenian Snowball-Earth episodes. The limited length
of the applied forcing (5000 yr) affects the statistics of extreme eruptions as well. Even longer model runs or ensembles, i.e. multiple simulations with the same sets of parameters, would help in assessing the thresholds more thoroughly. Nevertheless, our simulations are very well comparable with each other because they are based on the same pseudorandom time series (Figure 3); thus, we expect the systematic effects discussed above to be robust.

One should be cautious in taking the $CO_2$ thresholds literally, but they provide an estimate of the effects which one should
expect from permanent volcanism—ranging from an impact comparable to minor changes in the continental configuration and orbital geometry up to larger shifts than the most extreme orbital configurations.

## 6 Conclusions

Quantifying the threshold beyond which Earth falls into a fully ice-covered state is an important prerequisite for understanding the causes of the global (or near global) glaciations during the Neoproterozoic. Typically, this threshold is investigated in
terms of critical values of the solar constant and/or the atmospheric $CO_2$ concentration. Here, we study the influence of the continental configuration, Earth's orbital parameters, and perturbations from volcanic eruptions on the threshold for global glaciation. In the following, we briefly summarise our main findings.





**Figure 13.** Shift in $CO_2$ threshold for Snowball inception (red dots) for different scenarios of sequential volcanic eruptions (top row). Scenarios are specified at the top of each panel by indicating which parameter was changed in comparison to the reference scenario {$p = 0.1$, $\beta = 1$, $\xi = 0.3$, $u = -1.5\,\mathrm{W\,m^{-2}}$}. Histograms (bottom row) show the absolute frequency of eruptions with nominal peak forcings in bins of $1\,\mathrm{W\,m^{-2}}$. Red arrows point to the largest eruptions, specified by their peak forcings above the arrows. Blue and red rectangles indicate the expected ranges based on the effects of single global eruptions just below and above the largest events in the sequences; corresponding ranges in the histograms are grey. For the entire figure, blue colours signify perturbations on the coldest of the coexisting attractors, and red colours perturbations on the warmest attractors.

**Palaeogeography can have a significant influence on the Snowball-inception threshold.** The difference in thresholds is most pronounced for the two idealised continental configurations tested here: a widely dispersed distribution and a supercontinent centred at the South Pole. While the dispersed configuration is rather susceptible to global glaciation (with equilibrium thresholds between 190 and 250 ppm of $CO_2$ depending on the solar constant), the polar-supercontinent configuration proves to be intriguingly resistant (10–40 ppm). The thresholds of the two palaeogeographic reconstructions lie in between these extreme






cases (90–180 ppm depending on the solar constant) and differ by 20–40 ppm. The differences between the various continental configurations with respect to the Snowball-inception threshold can be primarily attributed to the influence of the land–ocean

distribution on absorbed solar radiation at Earth's surface, with only minor contributions from changes in the meridional heat transport and the greenhouse effect from atmospheric water vapour and clouds.

**Changes in orbital parameters lead to variations in the equilibrium threshold of up to 32 ppm.** These differences are thus of similar magnitude as the differences between the two palaeocontinental reconstructions. Orbital configurations with higher obliquity and lower eccentricity are generally colder and thus more susceptible with respect to global glaciation. The

dependence on eccentricity can be understood from the lower annual mean top-of-atmosphere insolation for less elongated orbits, while the cooling with increasing obliquity can be attributed to a shift of the sea-ice margin to lower latitudes caused by the more evenly distributed annual-mean solar flux, amplified by the ice–albedo feedback. In the context of many earlier modelling studies, it is noteworthy that an idealised configuration with an obliquity of $23.5°$ and a circular orbit (which is fairly close to modern orbital parameters) is rather susceptible to global glaciation.

**The effects of volcanic eruptions on the Snowball-inception threshold depend strongly on orbital geometry.** The differences are due to changes in the stability of attractors upon changing orbital parameters. Only very large eruptions with peak forcings of $-10\,\mathrm{W\,m^{-2}}$ or stronger can shift the threshold by a similar amount as changes in astronomical parameters. Compared to single eruptions with a globally-homogeneous forcing, latitudinally resolved scenarios show a dependence on the latitude of eruption. Tropical eruptions can have significantly larger impacts than high-latitude eruptions due to a different

response of surface temperatures. Simulations forced by 5000-yr-long sequences of volcanism allow at least two conclusions: first, the continuous noise-like perturbations can render formerly stable states unstable by connecting the basins of attraction. Second, the largest eruption in a time series mostly dictates the threshold similarly as single perturbations of the same magnitude; however, their effect can be amplified beyond the single perturbations by intense background noise from frequent volcanic eruptions.

**The sensitivity of Neoproterozoic Snowball inceptions is severely altered by the existence or nonexistence of an intermediate state of sea-ice extent.** This phenomenon is seen throughout all three tracks of simulations—considering continental configuration, orbital geometry, or volcanism—, where this 'Hadley state' with sea-ice margins around $30°$ in latitude may exists between the ubiquitous stable state with ice margins lying around $40°$ ('Ferrel state', Feulner et al., 2023) and the Snowball state. The Hadley state appears as an attractor accessible by quasistatic hysteresis simulations, thereby affecting the equilib-

rium threshold for Snowball inception, in three of our configurations: the Sturtian and Marinoan M+21 palaeogeographies with $23.5°$ obliquity and circular orbit, and the Sturtian L+13 continental reconstruction with $22°$ obliquity and circular orbit. Additionally, we find such an attractor hidden from hysteresis experiments but accessible by (volcanic) perturbations in the Sturtian L+13 configuration with $23.5°$ obliquity and circular orbit. For an aquaplanet setup, Feulner et al. (2023) argue that Hadley states are stabilised by large meridional temperature differences in the presence of a sufficiently high solar luminosity.

Despite the use of an oceanic general circulation model and a sea-ice scheme capturing dynamics, our model setup comes with limitations, two of which seem to be especially relevant for Snowball inceptions. First, the simplified atmosphere does not allow for changes in the Hadley-cell width, which is known to decrease under cold conditions (Frierson et al., 2007; Voigt,

2010), and might miscalculate the radiative transfer at low $CO_2$ concentrations. The properties of the observed coexisting Ferrel and Hadley attractors as well as the thresholds may be affected by these limitations. Nevertheless, we expect neither
the attractors to be sheer artifacts nor the thresholds to be largely erroneous as there exists evidence for similar stable states in more comprehensive models (Yang et al., 2012a, b, c) and the observed thresholds lie well within the ranges determined with atmosphere–ocean general circulation models (Feulner et al., 2023). Second, we cannot interactively simulate ice sheets. On the one hand, this means to neglect the dynamics of moving ice, which may have an influence on the critical conditions for full glaciation under equilibrium conditions (Hyde et al., 2000); on the other hand, the albedo of ice sheets may differ
significantly from snow on land and thereby could affect the thresholds. Furthermore, the absence of land ice alters the time scales of relaxation after perturbations. However, our perturbations act on time scales of years, which is too fast for significant changes in the land-ice extent.

In summary, we have shown that the distribution of continents, Earth's orbital parameters, and large volcanic eruptions can significantly shift the Snowball-inception threshold under Neoproterozoic boundary conditions. In terms of orbital geometry,
the parameter values representative of the modern situation and frequently used in modelling studies turn out to be among the more susceptible configurations with respect to global glaciation, indicating a potentially greater stability of the Neoproterozoic Earth system, at least when looking at the long-term average. On the other hand, large volcanic eruptions are an inevitable contributor to natural climate variability and can shift the glaciation threshold to higher $CO_2$ concentrations, implying that atmospheric $CO_2$ levels might not have had to sink as low as indicated by simulations ignoring their influence. In any case, it
is crucial to take these factors of natural climate variability into account in order to understand the global glaciations found in the geologic record.

*Code and data availability.* Model input and output files as well as the scripts used to generate the figures are available at the institutional repository of the Potsdam Institute for Climate Impact Research (Eberhard et al., 2023)[1]. The model source code will be made available upon request.

*Author contributions.* J.E., O.E.B., and G.F. designed the study; O.E.B. and G.F. prepared continental boundary-condition data with input from J.v.H. and J.U.L.B.; O.E.B. performed and analysed the model simulations on continental configurations with input from J.E. and G.F.; J.E. performed and analysed the model simulations on orbital geometry and volcanism with input from G.F.; S.P. provided technical assistance and compiled the data archive; J.E. prepared the figures; J.E., O.E.B., and G.F. wrote the paper with input from all co-authors.

*Competing interests.* The authors declare that they have no competing interests.

---

[1]preview URL: https://www.pik-potsdam.de/data/doi/10.5880/PIK.2023.002



*Acknowledgements.* This project has received funding from the European Research Council (ERC) under the European Union's Horizon 2020 research and innovation programme (Grant agreement No. 856555). The authors gratefully acknowledge the European Regional Development Fund (ERDF), the German Federal Ministry of Education and Research and the Land Brandenburg for supporting this project by providing resources on the high-performance computer system at the Potsdam Institute for Climate Impact Research.





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





**Appendix A: Supporting figures**

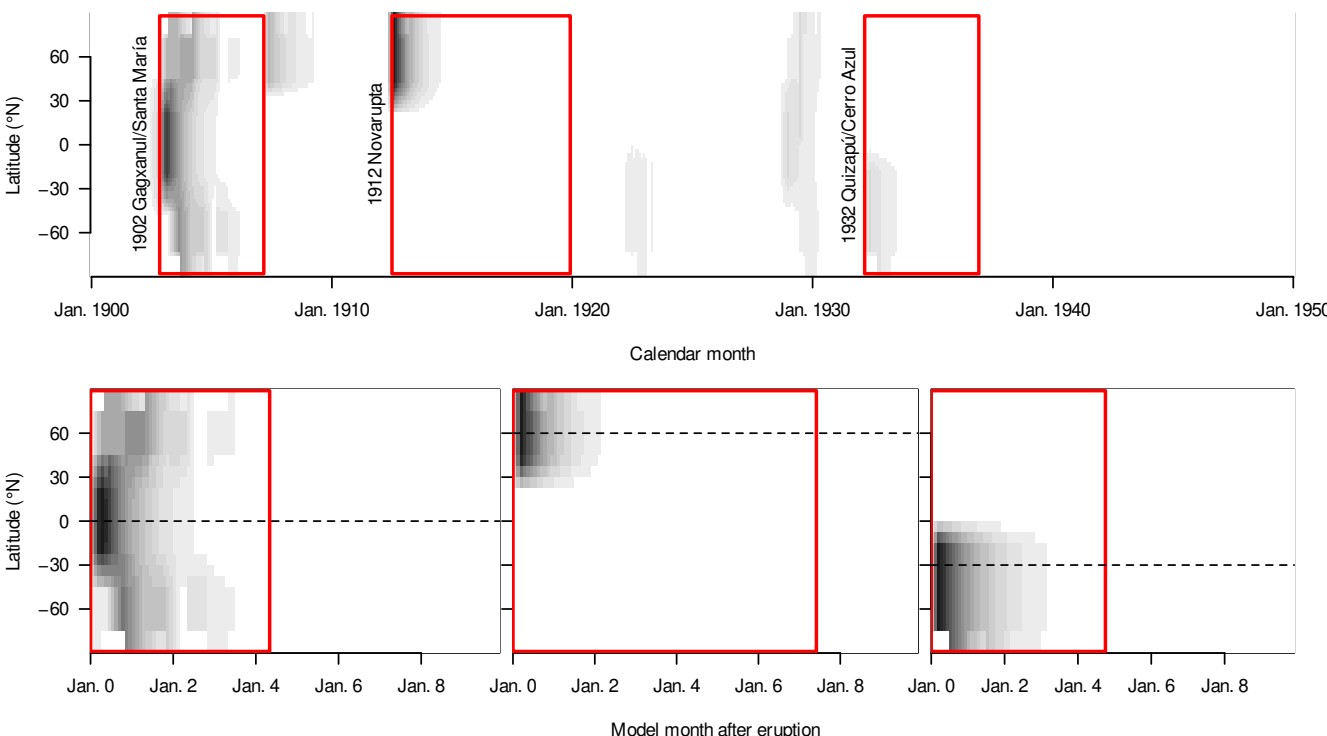

**Figure A1.** Scenarios of monthly- and zonally-resolved aerosol optical depth (AOD) following historical eruptions at $15°$ N, $58°$ N, and $36°$ S. The top panel shows the part of the original data (Ammann et al., 2003) containing the selected eruptions, the bottom panels show the first 10 yr of the spatially interpolated and temporally extended scenarios as used in the present study. Grey shading is proportional to AOD, but the correspondence between shading and AOD values differs between the panels.



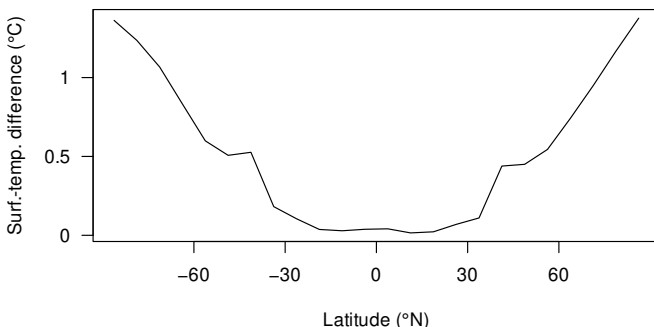

**Figure A2.** Difference in zonal-mean surface temperature between the estimation (2) and the actual model output for the Sturtian simulation with L+13 palaeogeography, a $CO_2$ concentration of 250 ppm, and an orbital geometry of $\{\varepsilon = 23.5°, e = 0\}$.

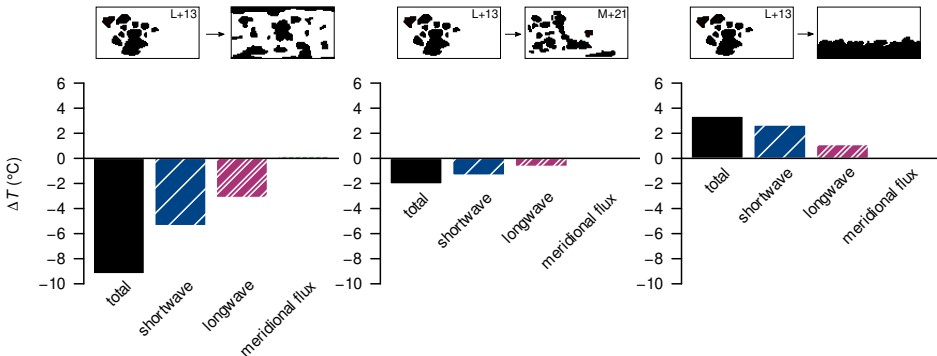

**Figure A3.** Globally averaged surface-temperature differences and contributions from net solar radiation ('shortwave'), emissivity ('long-wave'), and meridional heat-flux divergence ('meridional flux') for comparisons between the Sturtian L+13 continental configuration and the dispersed (left), Sturtian M+21 (middle), and polar-supercontinent configuration (right) at 720-Ma solar constant.

## Appendix B: Supporting table

**Figure A4.** Absolute difference in surface temperature $\Delta T$ (Ai–Ci), land fraction $\Delta f_{\text{land}}$ (Aii–Cii), land-snow and sea-ice fraction $\Delta f_{\text{cryo}}$ (Aiii–Ciii), cloud fraction $\Delta f_{\text{cloud}}$ (Aiv–Civ), and relative difference in column water vapour $\delta f_{\text{vap}} = \Delta \overline{\langle f_{\text{vap}} \rangle} / \overline{\langle f_{\text{vap}} \rangle}_{\text{ref}}$ (Av–Cv) between long-term-averaged attractor states upon changing the continental configuration for the Marinoan onset. Shown in Ai–Ci are the total temperature difference and differences due to changes in net solar radiation ('shortwave'), emissivity ('longwave'), and meridional heat-flux divergence ('meridional flux'). Differences are computed by subtracting the values of the Marinoan L+13 reconstruction from the values of the dispersed configuration (Ai–Av), the Marinoan M+21 (Bi–Bv), or the supercontinent configuration (Ci–Cv). The solar constant is always 1290 W m$^{-2}$ (650 Ma), the CO$_2$ concentration always 200 ppm.





**Table 4.** Overview of the simulations regarding the sensitivity of the $CO_2$ threshold for Snowball inception to single zonal volcanic perturbations. All simulations are run at a solar constant of $1283\,\mathrm{W\,m^{-2}}$, use the Sturtian L+13 palaeogeography, and have an orbital geometry of $22°$ obliquity and zero eccentricity.

| Initial partially ice-covered attractor | Nominal volcanic peak forcing ($\mathrm{W\,m^{-2}}$) | Latitude of eruption (° N) | $CO_2$ concentration (ppm) |
|---|---|---|---|
| Warmest | −2 | −58, −36, −15, 15, 36, 58 | 124 |
| | −5 | −58 | 128, 127 |
| | | −36 | 129, 128 |
| | | −15, 15, 36, 58 | 127, 126 |
| | −10 | −58, −36, −15, 15 | 133, 132 |
| | | 36 | 129, 128 |
| | | 58 | 127, 126 |
| | −20 | −58, −36, −15, 15, 36 | 133, 132 |
| | | 58 | 130, 129 |
| | −30 | −36, −15, 15, 36 | 133, 132 |
| | −40 | −36, −15, 15 | 136, 135 |
| | | 36 | 134, 133 |
| Coldest | −2 | −58, −36, −15, 15 | 125 |
| | | 36 | 127, 126 |
| | | 58 | 126, 125 |
| | −5 | −58 | 128, 127 |
| | | −36 | 129, 128 |
| | | −15, 15, 36, 58 | 127, 126 |
| | −10 | −58 | 134, 133 |
| | | −36, −15, 15 | 136, 135 |
| | | 36 | 129, 128 |
| | | 58 | 127, 126 |
| | −20 | −58 | 139, 138 |
| | | −36 | 144, 143 |
| | | −15 | 151, 150 |
| | | 15 | 153, 152 |
| | | 36 | 151, 150 |
| | | 58 | 130, 129 |
| | −30 | −36 | 154, 153 |
| | | −15 | 162, 161 |
| | | 15 | 164, 163 |
| | | 36 | 158, 157 |
| | −40 | −36 | 164, 163 |
| | | −15, 15 | 174, 173 |
| | | 36 | 151, 150 |



**Table 5.** Overview of the simulations regarding the sensitivity of the $CO_2$ threshold for Snowball inception to sequential volcanic perturbations. All simulations are run at a solar constant of $1283\,\mathrm{W\,m^{-2}}$, use the Sturtian L+13 palaeogeography, have an orbital geometry of $22°$ obliquity and zero eccentricity, and are initialised on the warmest attractor supported at the respective $CO_2$ concentrations.

| $p$ | $\beta$ | $u\,(\mathrm{W\,m^{-2}})$ | $CO_2$ concentration (ppm) |
|---|---|---|---|
| 0.1 | 1.0 | $-1.5$ | 155, 150 |
| 0.1 | 1.0 | 0.0 | 146, 145 |
| 0.1 | 1.0 | $-3.0$ | 165, 160 |
| 0.1 | 0.5 | $-1.5$ | 141, 140 |
| 0.1 | 1.5 | $-1.5$ | 170, 165 |
| 0.3 | 1.0 | $-1.5$ | 180, 175 |

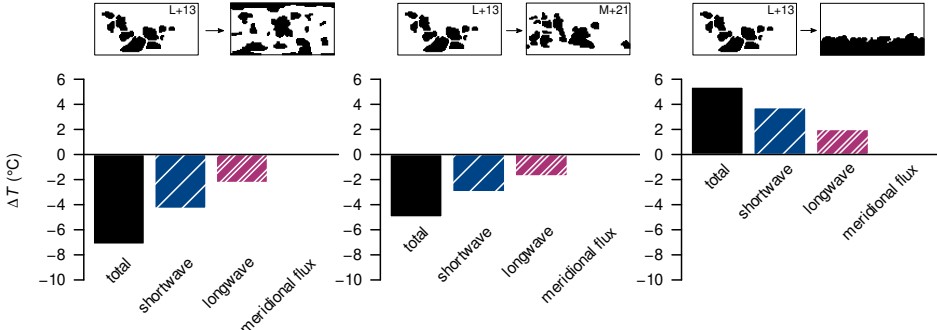

**Figure A5.** Globally averaged surface-temperature differences and contributions from net solar radiation ('shortwave'), emissivity ('longwave'), and meridional heat-flux divergence ('meridional flux') for comparisons between the Marinoan L+13 continental configuration and the dispersed (left), Marinoan M+21 (middle), and polar-supercontinent configuration (right) at 650-Ma solar constant.





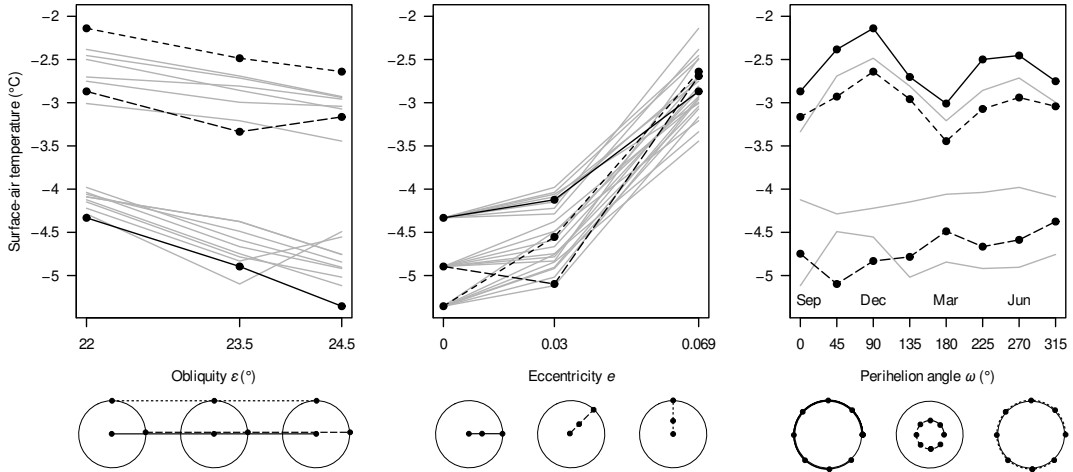

**Figure A6.** Isolated effects of varying the orbital parameters obliquity ($\varepsilon$, left), eccentricity ($e$, middle), and perihelion angle ($\omega$, right) on the long-term- and global-mean surface-air temperature, using the data shown in Figure 7. Below the diagrams, the selected 'transects' (black curves) in parameter space are shown by reference to the polar plots in Figure 7. All remaining curves drawn grey.

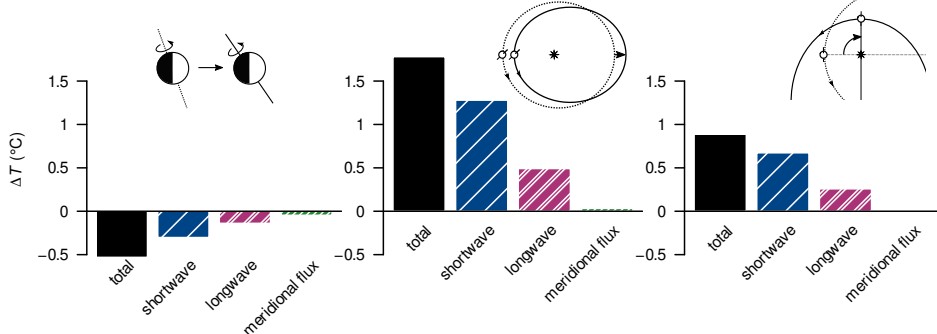

**Figure A7.** Globally averaged surface-temperature differences and contributions from net solar radiation ('shortwave'), emissivity ('longwave'), and meridional heat-flux divergence ('meridional flux').

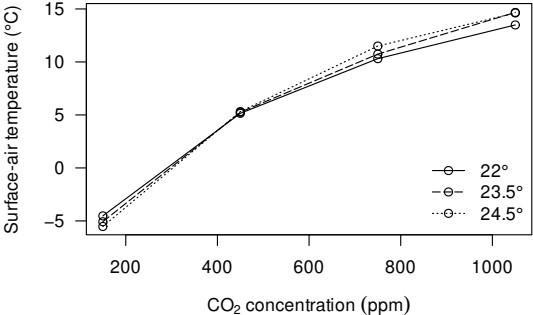

**Figure A8.** Long-term- and global-mean surface-air temperature for varied obliquity at different $CO_2$ concentrations.





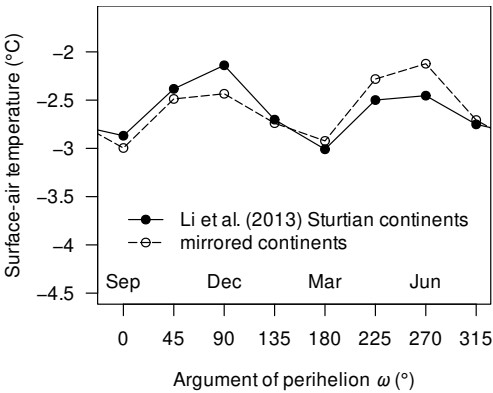

**Figure A9.** Long-term- and global-mean surface-air temperature for varying argument of perihelion $\omega$, once for the Sturtian L+13 palaeo-geography and once for the same configuration but with the continents mirrored at the equator.

**Table B1.** Long-term- and global-mean surface-air temperatures for different combinations of orbital parameters obliquity ($\varepsilon$), eccentricity ($e$), and argument of perihelion ($\omega$). Note that for $e = 0$, $\omega$ is not well-defined.

| | | $\omega =$ | | | | | | | |
|---|---|---|---|---|---|---|---|---|---|
| | | $0°$ | $45°$ | $90°$ | $135°$ | $180°$ | $225°$ | $270°$ | $315°$ |
| | $e = 0$ | $-4.33$ | | | | | | | |
| $\varepsilon = 22°$ | $e = 0.03$ | $-4.12$ | $-4.29$ | $-4.22$ | $-4.15$ | $-4.06$ | $-4.04$ | $-3.98$ | $-4.09$ |
| | $e = 0.069$ | $-2.87$ | $-2.38$ | $-2.14$ | $-2.70$ | $-3.01$ | $-2.50$ | $-2.45$ | $-2.75$ |
| | $e = 0$ | $-4.90$ | | | | | | | |
| $\varepsilon = 23.5°$ | $e = 0.03$ | $-4.75$ | $-5.10$ | $-4.83$ | $-4.78$ | $-4.49$ | $-4.67$ | $-4.59$ | $-4.38$ |
| | $e = 0.069$ | $-3.34$ | $-2.69$ | $-2.49$ | $-2.81$ | $-3.21$ | $-2.86$ | $-2.71$ | $-3.00$ |
| | $e = 0$ | $-5.35$ | | | | | | | |
| $\varepsilon = 24.5°$ | $e = 0.03$ | $-5.12$ | $-4.49$ | $-4.55$ | $-5.02$ | $-4.84$ | $-4.92$ | $-4.90$ | $-4.76$ |
| | $e = 0.069$ | $-3.16$ | $-2.93$ | $-2.64$ | $-2.96$ | $-3.44$ | $-3.07$ | $-2.94$ | $-3.04$ |





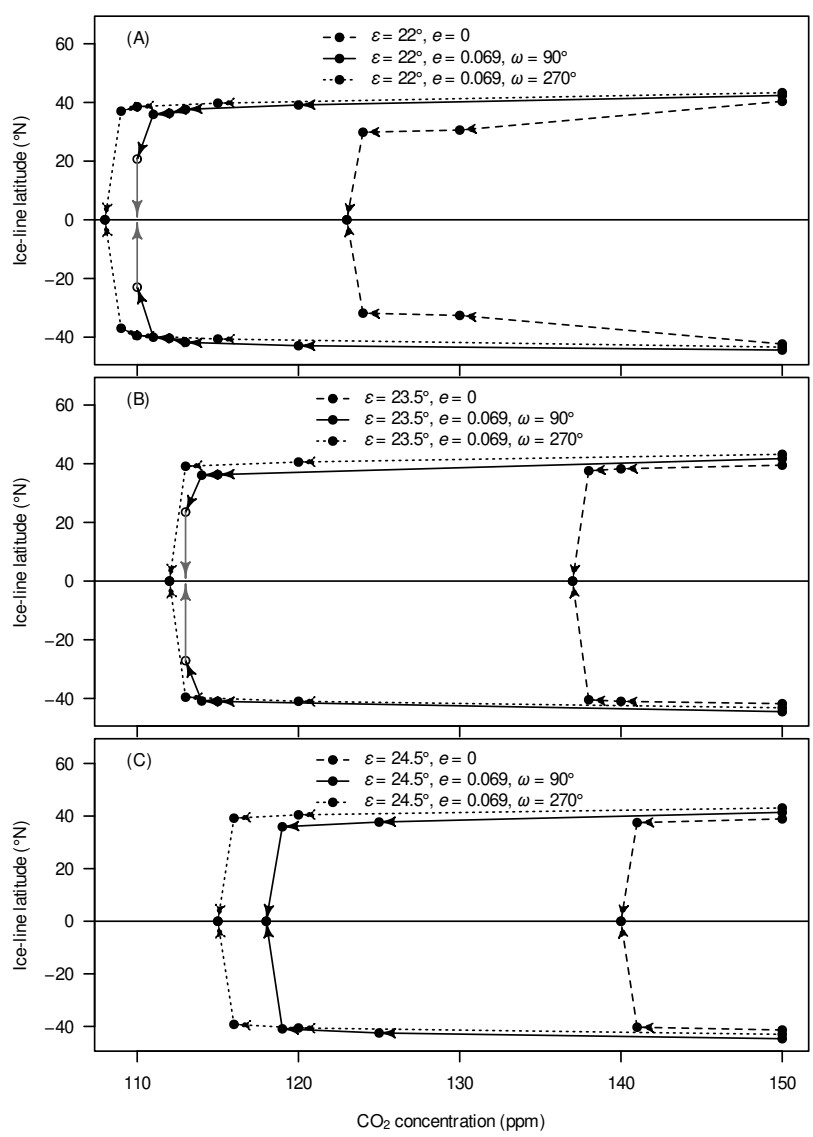

**Figure A10.** Long-term-mean ice-line latitudes, i.e. latitudes where the sea-ice concentration is 0.5, for the warmest attractors at different $CO_2$ concentrations between 150 ppm and the respective thresholds for nine different orbital geometries with $\varepsilon = 22°$ (A), 23.5° (B), and 24.5° (C).



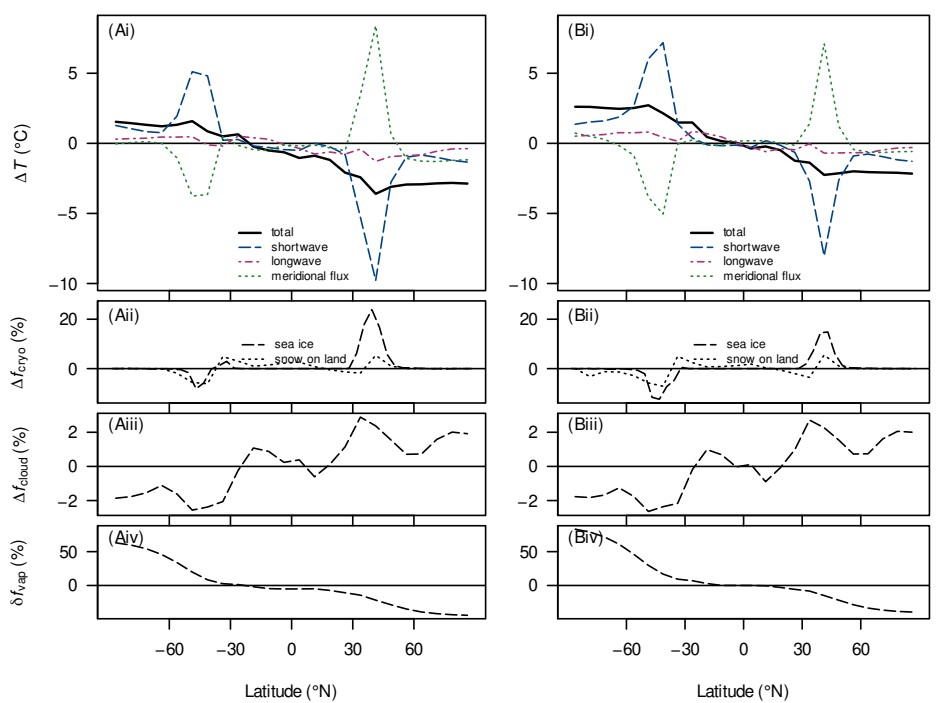

**Figure A11.** Absolute difference in surface temperature $\Delta T$ (Ai, Bi), land fraction $\Delta f_{\mathrm{land}}$ (Aii, Bii), land-snow and sea-ice fraction $\Delta f_{\mathrm{cryo}}$ (Aiii, Biii), cloud fraction $\Delta f_{\mathrm{cloud}}$ (Aiv, Biv), and relative difference in column water vapour $\delta f_{\mathrm{vap}} = \Delta \overline{\langle f_{\mathrm{vap}} \rangle} / \overline{\langle f_{\mathrm{vap}} \rangle}_{\mathrm{ref}}$ (Av, Bv) between long-term-averaged attractor states upon shifting the argument of perihelion as $\omega = 90° \to 270°$, once for a $CO_2$ concentration of 115 ppm (Ai–Av) and once for 112 ppm (Bi–Bv). Other orbital parameters are $\varepsilon = 22°$, $e = 0.069$.



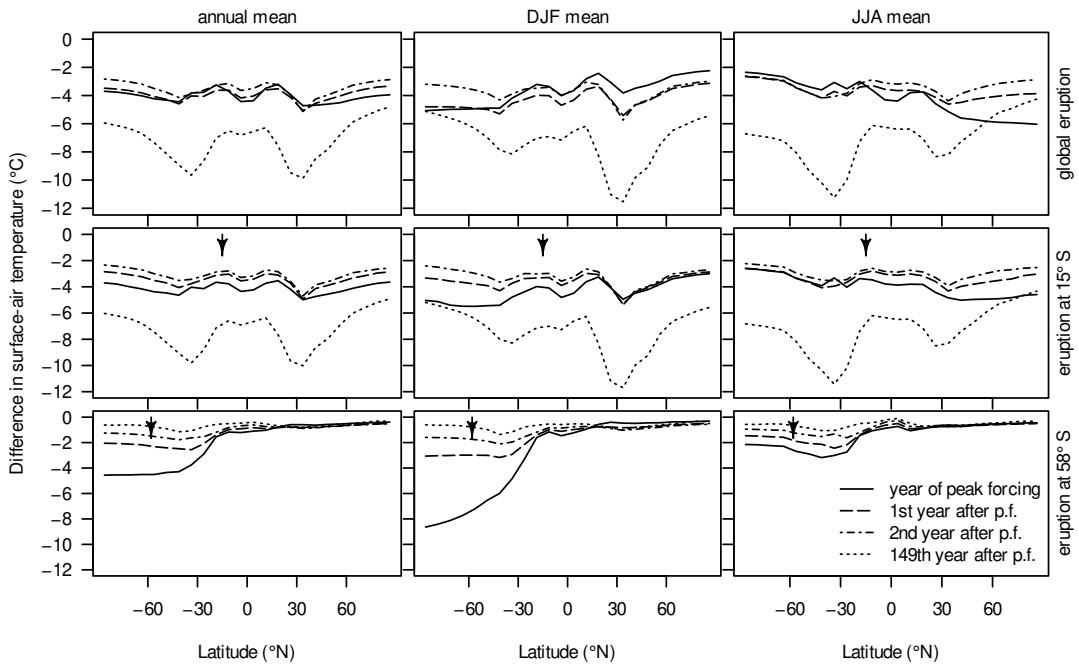

**Figure A12.** Annual (left column), December–January–February (DJF, middle column), and June–July–August (JJA, right column) means of zonal differences in surface-air temperature in selected years following volcanic eruptions with a nominal peak forcing of $-20\,\mathrm{W\,m^{-2}}$. Eruptions are applied as either globally-homogeneous perturbations (top row), zonal perturbations centred around $15°\,\mathrm{S}$ (middle row), or centred around $58°\,\mathrm{S}$ (bottom row). Differences are computed with respect to the last year before the eruption. Small arrows indicate the latitudes of eruption.