# Peer review of "Sensitivity of Neoproterozoic Snowball-Earth inceptions to continental configuration, orbital geometry, and volcanism"

_Climate of the Past, 2023_

## Author Response (AR1)

**cp-2023-38 Discussion:**
**Authors' response to referee comments**

September 13, 2023

In this document, we reproduce our answers to all three referees' comments and specify where and how we changed the manuscript. Referee comments are numbered and set in italic font, answers in upright running text, and changes listed as bullet points.

**RC1 by Yonggang Liu**

1. *A description of land surface albedo should be provided since it is important for understanding the why continental configuration can have such a large impact on the CO2-threshold.*

Thank you for this helpful suggestion. We agree that the albedo of the land surface is, similar to the cryosphere albedos, an important factor in the global energy balance. The albedo of land surface (bare soil) is 0.3 (infrared) and 0.15 (visible/ultraviolet radiation). We will add a sentence on the albedo of bare soil and, for completeness, ocean (having an albedo between 0.05 and 0.2) in the second paragraph of Section 2.1.

- We have added the following sentence to Section 2.1: "In the absence of ice or snow cover, bare-soil albedos are constant 0.3 (infrared) and 0.15 (visible/ultraviolet); ocean albedos assume values of 0.05–0.2 for clear-sky and 0.07 for overcast conditions, independent of the wavelength."

2. *L307-308. The meaning of this sentence is unclear.*

Thank you for pointing out this unclarity. In Section 5.2, we show that the response to single volcanic perturbations can differ—in some cases drastically—depending on the initial conditions, i.e. which attractor we let the perturbations act upon. With the sentence L306–308 we aim to clarify that this particular dependency on initial conditions is, in many cases, no longer given for sequential eruptions. The reason is that, starting from the warmest attractor, repeated perturbations often push the climate towards the coldest attractor in a matter of a few hundred or thousand years. Then, the response to the volcanic sequence is very similar to the case of initialising the model on the coldest (non-Snowball) attractor. We agree that understandability would benefit from a few improvements to the sentence, and we will modify the sentence to make it clearer.

- We have modified the sentence as follows (changes in bold): "The reason for choosing only the warmest attractors is that in many simulations the sequential perturbations **force** the model from

the warmest to the coldest attractor [...], in which cases the thresholds **would become the same as for simulations initialised on the coldest attractors**."

3. *'can attest to these observations' is ambiguous, it'd better be rewritten.*

Thank you for this comment. With this sentence we aim to point to the similarity between the results of the Sturtian and the Marinoan simulations. We will amend the sentence to remove the ambiguity.

- We have amended the sentence as follows: "[...] can attest to these observations. **For example, as with the Sturtian simulations, the dispersed and SH supercontinent configurations demonstrate extreme globally and annually averaged surface temperatures of –9.0 and 3.6 °C, respectively.**"

4. *L382-383. It is strange to me why the high-latitude continent has little snow on it in their model. Is it sublimated quickly? Similar puzzle appears at L424-426.*

Lines 382–383 refer to the dispersed continental configuration, featuring both a continent centred at the North Pole and a continent covering the South Pole, but allowing ocean bays to come close to the pole. In the model diagnostics, we see a significant difference in the annual-mean snow cover. The zonal mean of this can be seen in Figure 4Ai (which the referee's comment refers to). In the globally-resolved picture (not shown in the paper), snow cover and snowfall rates are especially high within some distance to the coastlines. Probably due to the different shapes, the northern continent has a rather large interior with little snowfall rates, while these regions are sparse and much smaller on the southern counterpart. Regarding snow sinks, both continents have no melting during most of the year except for a pronounced melting peak around their respective summer solstice (June in the north, December in the south). This melting event is larger on the northern than the southern continent. Sublimation does not happen on both continents at any time. Our conclusion is that the low snow cover on the norther continent is a result of low snowfall rates over its interior and melting events during summer.

For large parts of the polar supercontinent, we observe a similar situation as for the northern continent in the dispersed configuration (low snowfall rates and large distance to the coast).

- We have decided to not include this discussion in the manuscript, as the referee Dorian Abbot was not too happy about the paper's length and we consider the issue not essential for discussing our main results.

5. *The darkening effect of volcanic ash on snow and ice should be mentioned near the end manuscript. The ash, like dust, decreases the surface albedo of snow and ice when deposited and causes warming. The warming effect is significant when ice and snow are extensive on the globe, and has been demonstrated for both the snowball Earth (https://doi.org/10.1175/JCLI-D-20-0803.1) and the Last Glacial Maximum (https://doi.org/10.1029/2021GL096672). It is thus unclear to me whether a large volcanic eruption would be able to trigger a snowball Earth when such effect is considered in the model.*

Thank you very much for this comment. We think that you highlight an important point. Volcanic ash is not considered in our simulations and the model does not include any dust dynamics. We will add a sentence on this limitation in the model-description section (2.1) and discuss how this might affect our results in the conclusion.

- We have changed the last paragraph of Section 2.1 as follows: "In addition to the coarse spatial resolution, potential limitations of the model mostly result from the simplified atmosphere component, which statistically captures the large-scale atmospheric circulation patterns rather than simulating weather,  the absence of ice sheets**, and the missing effects of volcanic ash and dust**. Although [. . . ] without any land ice. Despite these **two limitations, the resulting equilibrium thresholds for Snowball inceptions** in the Neoproterozoic fall well within the range derived from more complex models (Feulner et al., 2023). **Additionally, however, the model does not include the transport and deposition of volcanic ash and dust, thereby also neglecting their darkening effects on snow.** We will discuss [. . . ]"

**RC2 by Dorian Abbot**

**Bigger Comments**:

1. ***Shorten paper****: I think the paper would be easier to digest if it were a little shorter. I would go through all the text and figures in the paper carefully and determine what is necessary to show and explain the main conclusions. I would move everything that is not necessary to supplemental material. This is just a suggestion though, and is not required for the paper to be published.*

Thank you for this suggestion. We see the difficulty of reading through a very long paper and were aware of this issue during preparation of the manuscript. During revision, we will try to shorten the manuscript wherever possible, especially in Sections 2.2, 2.3, and 5.2.

- During revision, we have found it difficult to significantly shorten the text without losing, from our point of view, relevant information. In particular, we have shortened a few parts of Sections 2.3, 3.2, and 5.2 by rephrasing, while adding a number of sentences due to other referee comments. The latter changes might just outweigh the former, and the paper is probably not shorter than before. Sorry.

2. ***Changes in CO2 threshold****: The radiative forcing of CO2 is approximately logarithmic. This means that reporting changes in the CO2 in ppm (linearly) does not make sense. A change of 30 ppm on a background concentration of 10 ppm is enormous, but a change of 30 ppm on a back- ground state of 300 ppm is small. All the references to changes in the CO2 threshold throughout the paper should therefore be changed to percentage change, rather than ppm change. So, for example, an increase from 100 ppm to 130 ppm should be reported as a 30% increase in the CO2 threshold, rather than a 30 ppm increase in the CO2 threshold.*

We are very grateful to you for pointing this out. It is a justified objection and we plan to change all occurrences in the paper accordingly.

- We have expressed all occurrences of changes in the $CO_2$ threshold as percentage change, relative to the lower threshold value, instead of ppm; we made one exception. The exception is the difference in the thresholds between idealised supercontinent and dispersed configuration, where the percentage change would have gone into the thousands. Instead, we decided to specify the change as a factor (e.g. "a factor of up to 19"). The modifications were applied to the Short Summary, the abstract, Sections 3.2, 4.2, 5.1, 5.2, 5.3, and 6, as well as Figures 10, 12, and 13.

**Smaller Comments**:

1. ***SW and meridional heat flux compensation****: In most of the plots examining differences between climate states using the 1D analysis framework, the shortwave and meridional flux terms compensate and nearly cancel out. It seems like this is something worth understanding and remarking on. Also, is there some identifiable process that determines their sum?*

Thank you for making this point. It is true that the meridional heat flux often seems to partly offset a certain warming or cooling caused by changes in the shortwave flux. Our working hypothesis is that the heat flux has this compensating tendency because, by and large, the atmospheric heat transport is described as a "macroturbulent" diffusive process in the atmosphere model POTSDAM-2 (see reference Petoukhov et al., 2000, in the manuscript). The observation that meridional heat flux and shortwave radiation appear to nearly cancel out is probably owing to the fact that the net shortwave radiation often is

the cause for local temperature changes and thus meridional temperature gradients, which the meridional flux then acts to partly offset; in comparison, the longwave contribution is less localised and therefore yields less pronounced changes in the divergence of the meridional heat flux.

Given that the longwave contribution is mostly reflecting local positive feedbacks involving water vapour and clouds, we consider it plausible that the sum of the shortwave and the meridional-heat-flux contributions can represent some kind of efficiency of the meridional heat flux in offsetting temperature differences: the larger it is, the less pronounced this tendency is.

The meridional heat transport only rarely determines the sign of the local temperature change: In most cases, the sum of shortwave and longwave contribution dominates whether there is cooling or warming— exceptions being the polar regions in Figures 4Ai and A4 Ai, and middle latitudes in Figure 8Ai, where the meridional heat transport 'tips the sign'. One should, however, also keep in mind that, by construction, the global sum of the heat-flux divergence is close to zero.

- We decided to not include this discussion in the paper and leave the public exchange as it is. The issue is indeed interesting and worth being discussed, but we do not consider it crucial for our main results.

2. **Continent map insets**: *Figures 4, 6, A3, A4, and A5 have small insets of the continental configurations. It would help the reader to add some marks for latitude, maybe every 30 degrees, since the latitude is referred to sometimes in the text.*

This is a good suggestion, which we are going to implement in the revised version of the manuscript.

- We have added marks to all insets.

3. **Obliquity effect**: *The authors attribute the cooling with increasing obliquity to "a shift of the sea-ice margin to lower latitudes caused by the more evenly distributed annual-mean solar flux." I wonder if another important factor is the background albedo distribution. Since the albedo is low at low latitudes and high at high latitudes, the insolation-weighted global-mean albedo will increase as the obliquity increases (in the range considered), which could also explain the observed cooling as the obliquity increases. This would be easy to miss using the 1D analysis framework because a local warming due to more shortave at high latitudes would actually lead to global cooling as more of the shortwave would be concentrated in a high albedo region.*

Thank you for this interesting comment. In fact, this hypothesis was discussed at a very early stage of manuscript writing. While we think that there may be a general cooling contribution from this effect for increasing obliquity in climate states with ice caps, there is an issue in attributing this effect: increasing the obliquity comes with an increase in seasonality, which is especially pronounced over the polar regions. Why a hemispheric ice extent increases or decreases under such an increased seasonality is difficult to assess under steady-state conditions, since many processes and feedbacks may play a role. In general, the ice extent can change, meaning that the insolation-weighted global-mean albedo would change already due to local albedo changes.

Furthermore, numerous studies (cited in L505 of the manuscript) report a warming under increased obliquity. If the cooling effect of insolation-weighted albedo plays a role there, it obviously will not dominate the other mechanisms.

- Due to the length of the manuscript, we prefer to not include this discussion in the paper. As

argued above, the effect is difficult to assess in our study and cannot explain the difference between responses to obliquity changes in warm and cold climate states.

**RC3 by Aiko Voigt**

1. *L168: What are the values for land and snow albedo?*

Values for the land-surface albedo (bare soil) are indeed missing in the manuscript. They are 0.3 (infrared) and 0.15 (visible/ultraviolet radiation). During revision, we will add this information, together with the open-ocean albedo, to the model description (Section 2.1). In that section (L125–128), snow albedos are already described.

- See response to comment 1 by Yonggang Liu.

2. *L194: I assume this means distinguishable in terms of the annual and seasonal distribution of solar irradiance at TOA?*

The argument of perihelion ($\omega$) indicates the time of year when Earth comes closest to the Sun. For a circular orbit, i.e. $e = 0$, the Sun–Earth distance is constant and $\omega$ is meaningless. In consequence, it is true that the annual and seasonal distribution of solar irradiance is indistinguishable.

- We have changed the sentence as follows: "Note that for $e = 0$, the different values of $\omega$ are **indistinguishable in terms of the annual and seasonal distribution of insolation**."

3. *L203: Out of curiosity, how many years per day can be simulated with the Climber model?*

On PIK's current high-performance compute cluster, 100 model years were simulated in 9 hours, i.e. around 267 model years per day on a single CPU.

- We did not add this information to the revised manuscript due to the length of the paper.

4. *Tab. 3: I was initially not sure what is meant by "initial partially ice-covered attractor" in the fourth column of Tab. 3. It becomes clear later in the text, but could also be explained here in the table caption.*

Thank you, this is a valid point. We will change the table caption accordingly.

- We have added the following sentence to the table caption: "The 'initial partially ice-covered attractor' refers to the conditions at which a particular simulation was initialised."

5. *L310: What is the thinking behind the approach to have a long volcanic forcing followed by no volcanic forcing?*

Thank you for pointing out the unclarity. The idea behind this simulation design was to ensure that we do not miss Snowball inceptions because the simulations might just end 'on their way' to a Snowball state. Since we know that it takes at least 1000 years for the model to relax to a partially ice-covered attractor, we consider it appropriate to give the model this amount of time without perturbations before judging whether or whether not it enters a Snowball state.

- We have amended the sentence in question as follows: "[. . . ]  we let the model run without perturbations for 1000 yr **in order to ensure that we do not miss an eruption-induced transition to a Snowball state**."

6. *L333: I am not sure I follow, should $\epsilon$ not be around 0.7, at least in a present-day like climate? It might be closer to 1 in a colder Snowball-like climate, of course. Reading my own PhD thesis again, I realized that the argument I made was not that $\epsilon \approx 1$ but that for typical values of $\epsilon$ of around 0.7, the fourth root is sufficiently close to 1. This is different from what the authors write here.*

You are right. We accidentally missed the fourth root in the expression, it should read $\epsilon^{1/4} \approx 1$.

- We have corrected the error.

7. *L350: Note: this is consistent with Voigt et al, 2011, where we also found that shifting higher albedo land to the tropics cools the planet by increasing the reflection of shortwave radiation, for the simple reason that insolation is higher in the tropics.*

Thank you for pointing this out. Yes, we see similar effects of increased albedo in the tropics, as discussed in the different subsections 3.1.1–3.1.3. Nevertheless it might help the reader to state this general relationship in a separate place, and we will add such a sentence in the paper.

- We have added the following sentence to Section 3.1: "Apart from the direct comparison between the reconstructions, there exists a negative correlation between tropical land area and global-mean surface temperature, which is consistent with previous studies (e.g. Voigt et al., 2011)."

8. *L394: Isn't this obvious? I.e., it describes the water vapor feedback, which of course is a feedback controlled by the temperature dependence of humidity.*

We agree that the existence of a feedback is quite obvious here. What was intended to say in L394/395 is that we do not expect *primary* reasons for a certain emissivity-induced warming/cooling other than the pure water-vapour feedback. However, as we discuss in the following three sentences (L395–399), there is an additional contribution from changing cloud cover.

- We decided to leave the sentence in question as it is.

9. *L460: I am not sure I understand what is meant with "requires a lower greenhouse influence to negate".*

In the sentence L459–461, we mean that in comparison to the Sturtian Snowball event, the higher insolation during the Marinoan yields, at global radiative equilibrium, also a higher rate of longwave emission. For a given $CO_2$ concentration, the global surface temperature is therefore higher than for the Sturtian conditions. Thus, in order to reduce the surface temperatures as far as necessary for a Snowball inception, the critical greenhouse-gas concentration has to be smaller (= "the greenhouse influence to negate has to be lower") than for Sturtian conditions. The wording may be too opaque for many to understand, and we will change it during revision.

- We have shortened the sentence as follows: "The increased resistance to glacial advance in the Marinoan simulations, as indicated by the colder climate states required, can be attributed to the higher solar irradiance  (Bahcall et al., 2001) **since the latter** results in a global net increase in surface temperatures Feulner2014." We simply deleted the unclear phrase because the argument works without it.

10. *L676: importance –> important*

We agree that replacing "of importance" by "important" would improve the readability. We will change the wording according to your suggestion.

- We have changed the wording as promised.

11. *Fig. 13: Why are there no y-labels for the absolute frequency plots?*

This is a valid objection. When preparing the manuscript, we thought that providing y-labels would not add much relevant information. Absolute frequencies, on the one hand, would depend on the length of the volcanic forcing, making them comparable within our study but not beyond it. Relative frequencies, on the other hand, would still (as absolute frequencies, too) depend on the size of bins subdividing the x-axis. However, for clarity of the figure, we will add absolute frequencies to the axis.

- We have added marks and values for the absolute frequencies.

12. *L705 and other parts in the manuscript: The effect is given in absolute numbers, i.e., in ppmv of CO2. Yet, the radiative forcing of CO2 is logarithmic and depends on the reference CO2. Thus, I think it would be helpful if the numbers could be put into context by either also mentioning the reference CO2 content or by converting them into a TOA radiative forcing.*

Thank you for this important suggestion. We agree with the problem of specifying $CO_2$ changes without reference. We would like to suggest to modify the manuscript in such a way that all differences in $CO_2$ concentration are indicated as percentage change.

- See response to Bigger Comment 2 by Dorian Abbot.

13. *L722: "similarly as single"? rephrase?*

"Similarly as" is used here synonymously with "in a similar way as". We agree that its meaning is not unambiguous here and will rephrase it during revision.

- We have rephrased the sentence accordingly.

14. *L727: Since some of the key results depend on it: how robust is the Hadley state? Is it a Climber-specific feature, or do the authors think it should also occur in other models, especially GCMs with actual atmospheric dynamics? This should be discussed I think.*

Thank you for this comment. It is indeed a relevant feature in the CLIMBER-3$\alpha$ model, but we think that the Hadley states have a reasonable physical foundation and similar states appear in a series of studies by Jun Yang and colleagues, see the discussion in Feulner, Bukenberger, and Petri (2023), *Earth Syst. Dynam.*, Section 4 (https://esd.copernicus.org/articles/14/533/2023/esd-14-533-2023.html). In L739–741, we write essentially the same and cite the three references.

- As explained above, we consider the discussion already complete in the initial manuscript and thus decided not to change it.

15. *L737: For the Snowball Hadley cell, please do not cite my PhD thesis but instead my corresponding paper in J. of Climate: Vogt, Held, Marotzke, 2011, https://doi.org/10.1175/JAS-D-11-083.1. Thanks!*

We will change the citation as you suggested.

• We have changed the citation.

16. *The title of Appendix B should come after Fig. A4.*

This is right; the wrong formatting was an unintentional result of the PDF compilation with pdfTeX.

• We have corrected the order.